# Unexplained repeated pregnancy loss is associated with altered perceptual and brain responses to men's body-odor

Liron Rozenkrantz[1,2†], Reut Weissgross[1,2†], Tali Weiss[1,2†], Inbal Ravreby[1,2], Idan Frumin[1,2], Sagit Shushan[1,2,3], Lior Gorodisky[1,2], Netta Reshef[1,2], Yael Holzman[1,2], Liron Pinchover[1,2], Yaara Endevelt-Shapira[1,2], Eva Mishor[1,2], Timna Soroka[1,2], Maya Finkel[1,2], Liav Tagania[1,2], Aharon Ravia[1,2], Ofer Perl[1,2], Edna Furman-Haran[2,4], Howard Carp[5], Noam Sobel[1,2]*

[1]Department of Neurobiology, Weizmann Institute of Science, Rehovot, Israel; [2]The Azrieli National Institute for Human Brain Imaging and Research, Rehovot, Israel; [3]Department of Otolaryngology & Head and Neck Surgery, Edith Wolfson Medical Center, Holon, Israel; [4]Life Sciences Core Facilities, Weizmann Institute of Science, Rehovot, Israel; [5]Department of Obstetrics & Gynecology, Sheba Medical Center, Tel Hashomer, Israel

**Abstract** Mammalian olfaction and reproduction are tightly linked, a link less explored in humans. Here, we asked whether human unexplained repeated pregnancy loss (uRPL) is associated with altered olfaction, and particularly altered olfactory responses to body-odor. We found that whereas most women with uRPL could identify the body-odor of their spouse, most control women could not. Moreover, women with uRPL rated the perceptual attributes of men's body-odor differently from controls. These pronounced differences were accompanied by an only modest albeit significant advantage in ordinary, non-body-odor-related olfaction in uRPL. Next, using structural and functional brain imaging, we found that in comparison to controls, most women with uRPL had smaller olfactory bulbs, yet increased hypothalamic response in association with men's body-odor. These findings combine to suggest altered olfactory perceptual and brain responses in women experiencing uRPL, particularly in relation to men's body-odor. Whether this link has any causal aspects to it remains to be explored.

*For correspondence:
noam.sobel@weizmann.ac.il

†These authors contributed
equally to this work

Competing interests: The
authors declare that no
competing interests exist.

Reviewing editor: Andreas T
Schaefer, The Francis Crick
Institute, United Kingdom

## Introduction

Mammalian reproduction is tightly linked to olfaction (*Parkes and Bruce, 1961*; *Wysocki and Lepri, 1991*; *Boehm et al., 2005*; *Petrulis, 2013*; *deCatanzaro, 2015*). This link has been extensively studied in rodents, where body-odors from adult males can lead to accelerated pubertal development in juvenile females (*Vandenbergh, 1967*), to synchronous estrous cycling in adult females (*Whitten, 1956*; *Whitten, 1958*), and to implantation failure (*Bruce, 1959*). Such social odors are typically transduced at the vomeronasal organ in the nose, and processed through the accessory olfactory system in the brain (*Dulac, 2000*; *Luo and Katz, 2004*), but there is also evidence for reproductive social odors acting through the main olfactory system as well (*Kang et al., 2009*; *Baum and Cherry, 2015*). Olfaction has also been linked to human reproduction. The best known, albeit controversial example of this is the phenomenon of menstrual synchrony. Here, women who cohabitate tend to have a synchronized menstrual cycle (*McClintock, 1971*), an effect mediated by women's body-odor (*Stern and McClintock, 1998*) (this effect has been challenged on statistical grounds (*Strassmann, 1999*), yet our view is that it holds). In a different set of related studies, the body-odor from breast-feeding women timed ovulation and menstruation in nulliparous women (*Jacob et al., 2004*),

and increased their sexual motivation (*Spencer et al., 2004*). In turn, men's body-odors were found to regulate menstrual cycle variability in women (*Cutler et al., 1986*; *Preti et al., 2003*). This is consistent with findings implying that women's preference for men's body-odors are menstrual cycle phase-dependent (*Havlicek et al., 2005*; *Rantala et al., 2006*). Moreover, maternal status was found to regulate neural responses to the body-odor of newborns (*Lundström et al., 2013*). Body-odors also impact reproduction relevant mechanisms in men. For example, attractiveness ratings applied by men to women's body-odors are menstrual-phase specific (*Kuukasjarvi, 2004*), with men rating women's axillary and vaginal odors as most pleasant around ovulation (*Doty et al., 1975*; *Singh and Bronstad, 2001*; *Gildersleeve et al., 2012*). In addition, sniffing emotional tears collected from women donors reduced levels of testosterone (*Gelstein et al., 2011*; *Oh et al., 2012*) and ensuing arousal in men (*Gelstein et al., 2011*).

Given that the olfactory system is associated with human reproduction, we hypothesized that it may also be related to disorders of human reproduction. One such human reproductive disorder is unexplained repeated pregnancy loss (uRPL). A remarkable ~50% of all human conceptions, and ~15% of documented human pregnancies, end in spontaneous miscarriage (*Rai and Regan, 2006*). Only a limited portion of these miscarriages, however, can be explained (*Clifford et al., 1994*; *Stephenson, 1996*; *Jaslow et al., 2010*), suggesting the presence of major yet-unidentified human-miscarriage-related factors. Given the general link between olfaction and reproduction, we set out to test the hypothesis that olfaction is altered in uRPL. Moreover, because the role of olfaction in reproduction is typically related to body-odors, we used these as a primary target in our investigation.

## Results

### Whereas control women cannot identify their spouse by smell, women with uRPL can

A key olfactory ability investigated in the context of reproduction is the ability to identify mates by smell (*Leinders-Zufall et al., 2004*; *Brennan and Zufall, 2006*). To characterize olfactory spouse-recognition in humans, we tested 33 women with uRPL (mean age = 34.7 ± 6.4, mean number of miscarriages = 4.2) and 33 matched controls (mean age = 35.6 ± 4.1, never experienced miscarriage). We used a three-alternative forced- choice (3AFC) paradigm where on each trial the participant was asked to identify her spouse from three alternatives (delivered in specially designed body-odor sniff-jars, see *Figure 1a*); one containing the body-odor of her spouse, and two containing distractor odors; one from a *non-spouse man* and one *Blank* (carrier alone). Previous results on spouse odor identification in the general population are mixed, with two studies reporting ~33% group-level performance (*Hold and Schleidt, 1977*; *Mahmut et al., 2019*) but one study reporting ~74% performance (*Lundström and Jones-Gotman, 2009*). Here, consistent with the two former studies, as a group, control women were unable to identify their spouse above chance levels (chance = 33.3%, mean control = 28 ± 34%, Shapiro-Wilk test of normality (SW) = 0.77, p<0.001, difference from chance: Wilcoxon W = 206, p=0.18, effect size estimated by rank biserial correlation (RBC) = 1.17 (parametric comparison: t(32) = 0.88, p=0.39, Cohen's d = 0.15)). In contrast, remarkably, uRPL women were able to identify their spouse by body-odor alone, at a level that was both significantly higher than expected by chance (chance = 33.3%, mean uRPL = 57 ± 41%, SW = 0.81, p<0.001), difference from chance: Wilcoxon W = 440, p=0.004, RBC = 1.71 (parametric comparison: t(32) = 3.32, p=0.002, Cohen's d = 0.58), and two-fold that of controls (Mann-Whitney U = 332.5, p=0.005, RBC = 0.39, parametric comparison: t(64) = 3.12, p=0.003, Cohen's d = 0.77) (*Figure 1b*). To further verify that this result was not a chance effects, we conducted a bootstrap test. We randomly shuffled the assignment of the data to uRPL or Control 10,000 times, and reconducted the analysis. We observe that the probability of obtaining our results coincidentally is 2 in 1000, further suggesting that this result was not a chance event (*Figure 1c*). We note that over time, we modified the task to use six rather than four 3AFC trials per participant (and three participants had only two 3AFC trials). However, this had no impact on group results, and when we reanalyze the entire dataset restricting to either four 3AFC trials per participant or two 3AFC trials per participant, the uRPL group remained significantly better than the control group in all cases (all Mann-Whitney U > 386.5, all p<0.026, all RBC > 0.29, parametric comparison: all t(64) > 2.4, all p<0.018, all Cohen's d > 0.6). In

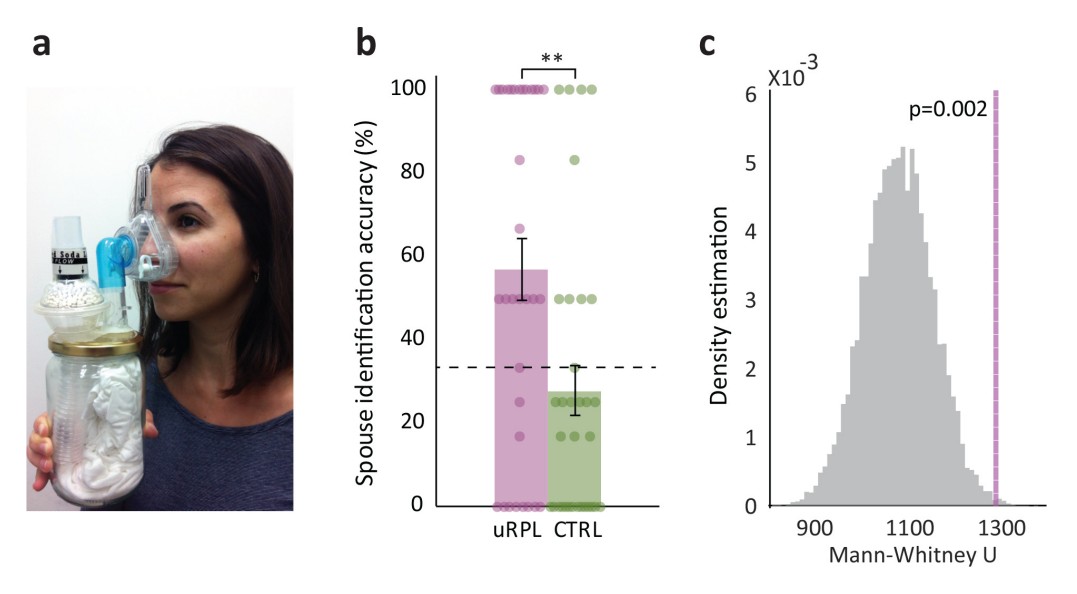

**Figure 1.** Women with uRPL can identify their spouse by smell. (**a**) A custom-designed Shirt Sniffing Device (SSD) to standardize body-odor sampling. The SSD consists of a glass jar containing the T-shirt, with an air intake port via soda lime filter, and air sampling port via one-way flap valve into individual-use airtight nose mask. This arrangement assured that environmental odors didn't contaminate the sample during the sampling process. The recognizable person in the figure is a co-author and not a participant. (**b**) Performance at 3AFC identification test (n = 66). uRPL women in purple, control women in green. Bar graphs depict group means, each dot represents a participant, dots are jittered to prevent overlay, error bars are s.e.m, **p<0.01. Black dashed line indicates chance level. (**c**) Bootstrap analysis. Gray lines represent the 10,000 repetitions; the purple line represents the actual uRPL-control difference value.

The online version of this article includes the following source data for figure 1:

**Source data 1.** All spouse identification values.

other words, an increased tendency for miscarriage in women is associated with better identification of spouse body-odor at the group level.

## Women with uRPL have otherwise only marginally better olfaction than controls

In order to determine whether this advantage at identifying spouse body-odor was a reflection of a generally better sense of smell, we tested olfactory performance at several additional tasks. To test ordinary odor identification, 38 women with uRPL and 38 matched controls conducted a four-alternative forced-choice identification test using 20 every-day odorants such as *soap* and *peanuts*. We observed only a trend toward better identification in uRPL women than in controls (mean uRPL = 87.6% ± 7.2, Controls = 84.6% ± 7.3, uRPL SW = 0.91, p=0.005, control SW = 0.92, p=0.009, Mann-Whitney U = 560.5, p=0.087, RBC = 0.22 (parametric comparison: t(74) = 1.82, p=0.073, Cohen's d = 0.42)) (*Figure 2a*). Subsequently, 38 women with uRPL and 38 controls were tested for detection of three odorant monomolecules that have often been studied in the context of human social chemosignaling (*Jacob and McClintock, 2000*). The monomolecules were Androstenone (ANN), Androstadienone (AND), and Estratetraenol (EST). We used each odorant at a fixed concentration, and tested its discrimination from blank using a 3AFC test. To account for the non-normal distribution of the data, a linear mixed-model with fixed effects of Group and Odor, with random intercept per participant and a random slope for odor within participant, revealed a main effect of Odorant (F(2,143) = 17.1, p<0. 0001), a modest effect of Group (F(1,74) = 3.97, p=0.05) and no interaction (F(2,143) = 0.27, p=0.77) (for comparison, a parametric repeated-measures analysis of variance (ANOVA) with factors of Group (RPL/Control) and Odorant (ANN/AND/EST) revealed a main effect of Odorant (F(2,148) = 17.5, p=1.5e$^{-7}$), but only a trend toward a main effect of Group (F(1,74) = 3.28, p=0.074), and no interaction (F(2,148)=0.31, p=0.74) (*Figure 2b*)). The main effect of Odorant reflected that at these particular concentrations, EST was harder to detect than both AND and ANN

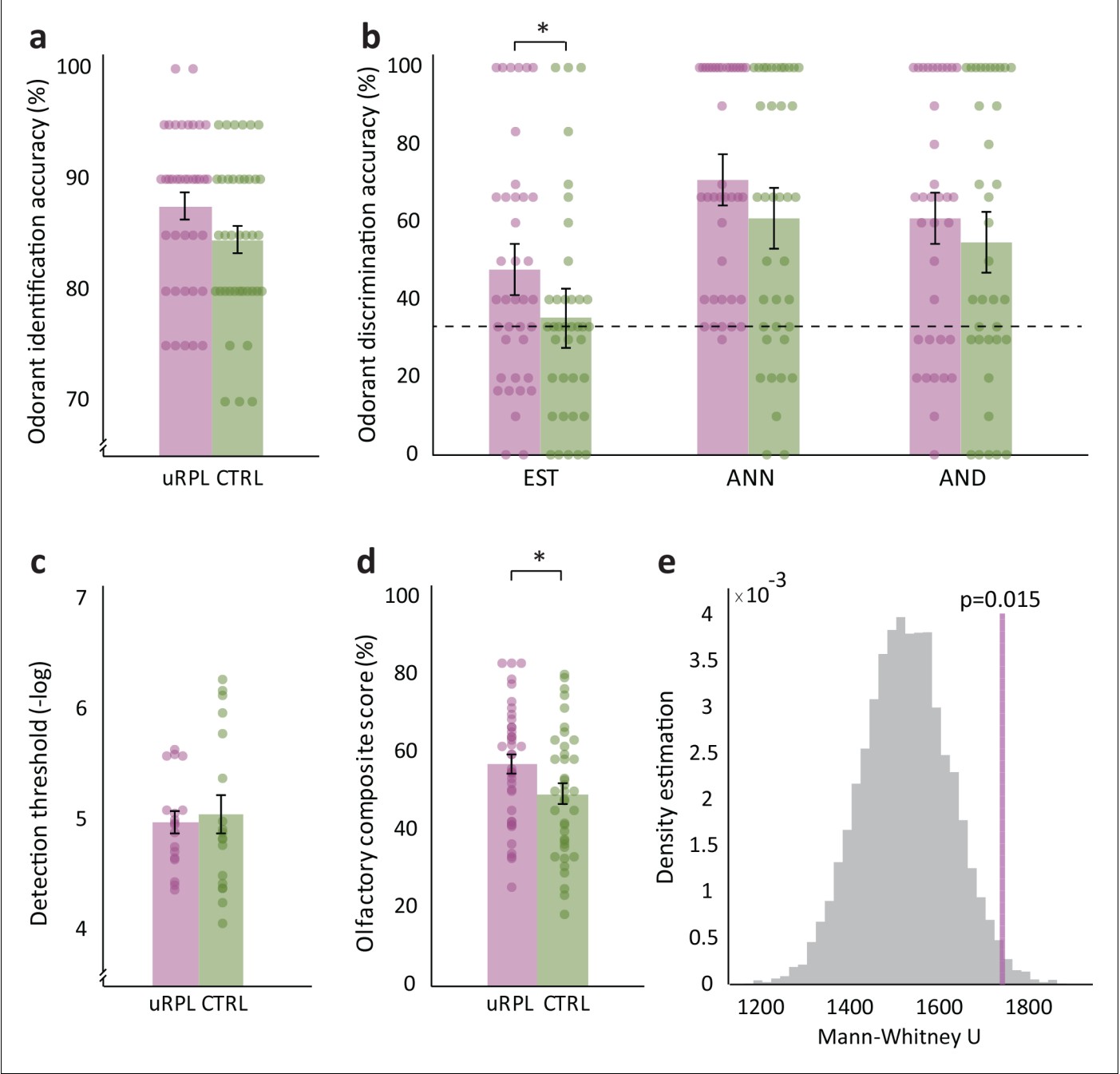

**Figure 2.** Women with uRPL have slightly better olfaction than controls. uRPL (purple) and control (green) women were tested for various olfactory facets. (a) Percent accuracy at every-day odorant identification (n = 76). (b) Percent accuracy at monomolecule discrimination (EST, ANN, AND) (n = 76). (c) DMTS threshold (n = 36). (d) A composite score of identification, discrimination and threshold (n = 78). (e) Bootstrap test of 'd', 10,000 repetitions, the purple line represents the actual uRPL-control difference value. Bar graphs depict means, each dot represents a participant, dots are jittered to prevent overlay, error bars are s.e.m. *p<0.05. Black dashed line indicates chance level.
The online version of this article includes the following source data for figure 2:

**Source data 1.** All odourant identification, discrimination, and detection values, including composite score.

(mean EST = $41.8 \pm 29.4$, mean AND = $58.1 \pm 34.6$, mean ANN = $66.2 \pm 31$: AND vs. ANN Wilcoxon W = 858, p=0.068, RBC = 0.29 (parametric comparison: t(70) = 1.7, p=0.093, Cohen's d = 0.2); AND vs EST Wilcoxon W = 451, p=$6.4e^{-4}$, RBC = 0.51 (parametric comparison: t(70) = 3.5, p=$8.2e^{-4}$, Cohen's d = 0.42); ANN vs. EST: Wilcoxon W = 237, p=$6e^{-7}$, RBC = 0.74 (parametric comparison: t

(75) = 5.86, p=$1.2e^{-7}$, Cohen's d = 0.67) (*Figure 2b*). The trend toward a main effect of Group reflected that uRPL participants again performed slightly better than controls, but this effect was near-significant for the odorant EST alone (EST: uRPL mean = 48.1 ± 30.3%, Control mean = 35.4 ± 27.4%, Mann-Whitney U = 544.5, p=0.065, RBC = 0.25 (parametric comparison: t(74) = 1.9, p=0.06, Cohen's d = 0.44); AND: uRPL mean = 61.1 ± 32.5%, Control mean = 55.1 ± 36.8%, Mann-Whitney U = 585, p=0.6, RBC = 0.07 (parametric comparison: t(69) = 0.73, p=0.47, Cohen's d = 0.17); ANN: uRPL mean = 71.1% ±27.1%, Control mean = 61.3 ± 34%, Mann-Whitney U = 587.5, p=0.15, RBC = 0.19 (parametric comparison: t(74) = 1.38, p=0.17, Cohen's d = 0.32) (*Figure 2b*). Finally, a group of 18 women with uRPL and 18 controls were also available for a lengthy seven-reversal detection-threshold staircase paradigm using the alliaceous odorant dimethyl trisulfide (DMTS). We observed no significant difference between controls and uRPL (uRPL threshold, minus log concentration: 4.98 ± 0.41, control threshold: 5.06 ± 0.72, uRPL SW = 0.92, p=0.13, control SW = 0.9, p=0.052, t(34) = 0.42, p=0.68, Cohen's d = 0.14) (*Figure 2c*). In reviewing the above results at olfactory identification, discrimination, and detection, women with uRPL repeatedly scored slightly better than controls, but this difference was not significant. For added power and sensitivity, in a final comparison we combined the above results, generating a composite olfaction score for each participant (reflecting her available scores at identification, discrimination, and detection) ranging between 0 and 100 (see Methods). This enabled a comparison between 39 women with uRPL and 39 matched controls. We observed a small but significant difference between the two groups (mean composite uRPL = 57.13% ± 15.23, Controls = 49.46% ± 16.14, uRPL SW = 0.97, p=0.49, control SW = 0.98, p=0.67, independent t-test t(76) = 2.16, p=0.034, Cohen's d = 0.49) (*Figure 2d*). To verify that this was not a chance event, we again conducted a 10,000-repetion bootstrap like before, and observe that the probability of obtaining this result by chance is less than 2 in 100 (*Figure 2e*). We further tested for any relationship between this composite score and the overwhelmingly better olfactory identification of spouse in uRPL versus controls. We observed no correlation across all participants, but a trend in uRPL alone (uRPL group (n = 33): Spearman Rho = −0.33, p=0.06; Control group (n = 29): R = 0.06, p=0.76; Both groups together: R = −0.02, p=0.88). In conclusion, at the group level, women with uRPL were two-fold better than control women at identifying their spouse by smell, and marginally but significantly better than control women at ordinary olfactory tasks.

## Women with uRPL have altered perception of men's body-odor

Given that the advantage at identifying spouse by smell was unrelated to general olfactory abilities, we next asked whether it was reflected in any explicit perceptual olfactory attributes in body-odor. A cohort of 18 women with uRPL and 18 matched controls rated men's body-odors on four traits: Intensity; Pleasantness; Sexual attraction; and Fertility. Each woman blindly (but see later note) rated stimuli of three kinds: *Blank* (carrier alone), *a non-spouse man*, and their actual *Spouse*. To account for the non-normal distribution of the data, a linear mixed-model with fixed effects of Group, Odor and Descriptor, and random effects of participant, revealed a main effect of Descriptor (F(3,374) = 9.67, p=$3.69e^{-6}$), and significant interactions of Odor x Descriptor (F(6,374) = 5.69, p=$1.13e^{-5}$) and Group x Odor (F(2,374) = 7.77, p=$4.95e^{-4}$) (parametric comparison: a multivariate repeated measures ANOVA with factors of Group (uRPL/Control), Odor (Blank/Non-Spouse/Spouse) and Descriptor (Intensity/Pleasantness/Sexual attraction/Fertility) revealed a main effect of Descriptor (F(3,102) = 12.4, p=$5.6e^{-7}$), a significant interaction of Odor x Descriptor (F(6,204) = 8.27, p=$5.1e^{-8}$) and a significant interaction of Odor x Group (F(2,68) = 3.43, p=0.038) (*Figure 3a*)). The main effect of Descriptor is uninteresting in that it merely reflected that descriptors were applied differently (mean Intensity: 0.49 ± 0.21, mean Pleasantness: 0.42 ± 0.14, mean Sexual attraction: 0.33 ± 0.15, mean Fertility: 0.38 ± 0.15; Wilcoxon pairwise comparisons except intensity-pleasantness: all W > 475, all p<0.026, intensity-pleasantness Wilcoxon W = 431, p=0.127 (parametric comparison, all t(35) > 2.12, all p<0.041, all Cohen's d > 0.35) (*Figure 3—figure supplement 1*). The Odor x Descriptor interaction was carried by intensity ratings alone (F(2,68) = 16.2, p=$1.72e^{-6}$; all other descriptors: all F(2,68) < 2.05, all p>0.13). This intensity difference was carried solely by differences from Blank, which was, unsurprisingly, less intense than all other otherwise (and importantly) equally-intense stimuli (mean Intensity ratings: Blank:=0.31 ± 0.26, Non-Spouse = 0.58 ± 0.23, Spouse:=0.59 ± 0.35. Blank vs. Non-Spouse and Spouse: both Wilcoxon W < 98, p<0.0001, RBC >0.7 (parametric comparison: Both t(35) > 4.7, both p<$3.8e^{-5}$, both Cohen's d > 0.77); Non-Spouse vs Spouse: Wilcoxon W < 319, p=0.83, RBC = 0.04 (parametric comparison: t(35) = 0.15, p=0.88, Cohen's d = 0.03)

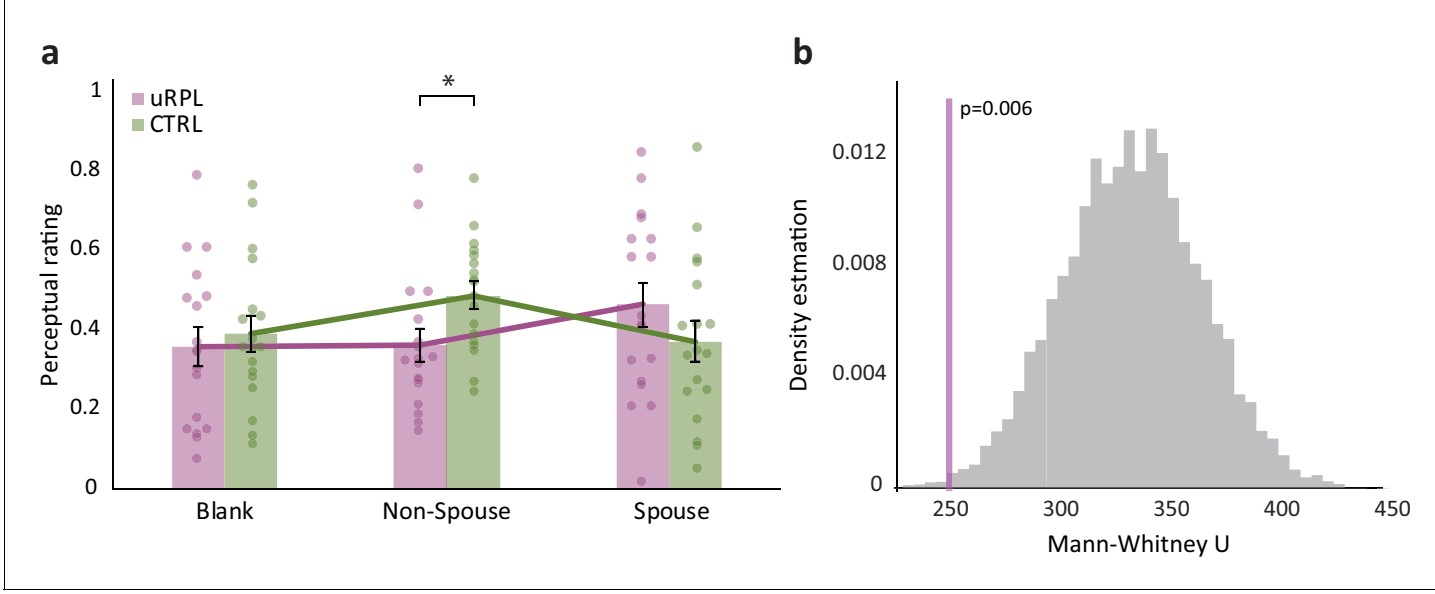

**Figure 3.** Women with uRPL have altered perception of men's body-odor. (a) uRPL (purple) and control (green) women ratings of non-spouse men body-odor, score combines ratings of pleasantness, sexual attraction, intensity and fertility attributed to the odor (see separate ratings in *Figure 3— figure supplement 1*). n = 36. Bar graphs depict means, each dot represents a participant, dots are jittered to prevent overlay, error bars are s.e.m. *p<0.05. (b) Bootstrap test of the non-spouse result, 10,000 repetitions, the purple line represents the actual uRPL-control difference value. The online version of this article includes the following source data and figure supplement(s) for figure 3:

**Source data 1.** All perceptual ratings of blank, non-spouse, and spouse.
**Figure supplement 1.** Altered perception of men's body-odor in uRPL.

(*Figure 3—figure supplement 1*). As to the Odor x Group interaction, whereas uRPL and Control women rated Blank and Spouse the same (uRPL Blank:=0.36 ± 0.2; Control Blank:=0.39 ± 0.18, Mann-Whitney U = 173, p=0.74, RBC = 0.07 (parametric comparison: t(34) = 0.49, p=0.63, Cohen's d = 0.16); uRPL Spouse = 0.46 ± 0.23; Control Spouse = 0.37 ± 0.21, Mann- Whitney U = 121, p = 0.2, RBC = 0.25 (parametric comparison: t(34) = 1.25, p=0.22, Cohen's d = 0.42), Non-Spouse was rated lower by uRPL than by Controls (uRPL = 0.36 ± 0.18; Controls = 0.49 ± 0.14, Mann-Whitney U = 244, p=0.009, RBC = 0.51 (parametric comparison: t(34) = 2.35, p=0.025, Cohen's d = 0.78) (*Figure 3a*). This effect that materialized in the combined descriptors was mostly carried by a difference is rated fertility attributed to the body-odors (*Figure 3—figure supplement 1*). In the spouse condition, uRPL and control women smelled different stimuli (each woman smelled her own spouse), so this was not an optimal condition for estimating differences in olfactory perception across these groups (we acknowledge that this condition adds the tantalizing possibility that in addition to altered olfaction in women with uRPL, men in uRPL relationships may have a unique body-odor, yet we do not have statistical power for this separate question within the current study). However, in the *non-spouse man* condition, both groups are smelling common stimuli, and we observed a significant difference in perception. Again, to verify that this result with non-spouse odors was not a chance event, we conducted a 10,000-repetion bootstrap like before, and observe that the probability of obtaining this result by chance is less than 6 in 1000 (*Figure 3b*). In other words, we observed a difference in the overall explicit perceptual ratings applied to independent men's body-odor by women with uRPL versus controls.

## Women with uRPL have smaller olfactory bulbs and shallower olfactory sulci

Results to this point imply an altered sense of smell in women with uRPL, specifically in relation to men's body-odor. To probe for brain correlates of this difference, we used magnetic resonance imaging (MRI) to scan 23 women with uRPL and 23 controls. To compare brain structure, we applied both a targeted and a hypothesis-free approach. In the targeted investigation, we compared brain

volume in the two primary regions of interest where brain structure has previously been related to olfactory function, namely the olfactory bulbs (*Buschhüter et al., 2008*) and the olfactory sulci (*Rombaux et al., 2010*). We observed that women with uRPL have significantly smaller olfactory bulbs (all SW >0.94, all p>0.2; Right bulb volume: uRPL: 45.9 ± 12 mm$^3$, control: 55.4 ± 9.7 mm$^3$, t (44) = 2.96, p=0.005, Cohen's d = 0.87; Left bulb volume: uRPL: 46.4 ± 8.9 mm$^3$, control: 55.7 ± 9.4 mm$^3$, t(44) = 3.45, p=0.001, Cohen's d = 1.02; Average bulb volume: uRPL: 46.1 ± 9.9 mm$^3$, control: 55.5 ± 9 mm$^3$, t(44) = 3.37, p=0.0016, Cohen's d = 0.99) (*Figure 4a*). To verify that this was not a chance event, we again conducted a 10,000-repetion bootstrap like before, and observe that the probability of obtaining this result by chance is less than 1 in 1000 (*Figure 4b*). Women with uRPL also had significantly shallower olfactory sulci (Right olfactory sulcus depth did not reach significance: uRPL: 6.4 ± 1.1 mm, control: 7.1 ± 1.4 mm, t(44) = 1.68, p=0.1, Cohen's d = 0.5; Left olfactory sulcus depth: uRPL: 6.1 ± 1.4 mm, control: 7 ± 1.4 mm, t(44) = 2.22, p=0.03, Cohen's d = 0.66; Average olfactory sulcus depth: uRPL: 6.3 ± 1.1 mm, control: 7 ± 1.3 mm, t(44) = 2.12, p=0.04, Cohen's d = 0.62) (*Figure 4c*). To verify that this was not a chance event, we again conducted a 10,000-repetion bootstrap like before, and observe that the probability of obtaining this result by chance is less than 2 in 100 (*Figure 4d*). To verify that these anatomical differences were not merely a reflection of smaller heads/brains, we conducted the same analysis on Total Intracranial Volume (TIV), and observe no difference between uRPL women and controls (TIV: uRPL mean: 1366 ± 134 cm$^3$, Control mean: 1390 ± 93 cm$^3$, uRPL SW = 0.95, p=0.31, control SW = 0.94, p=0.2, t(43) = 0.69, p=0.5, Cohen's d = 0.21). As olfactory bulb volume has been positively associated with olfactory performance (*Buschhüter et al., 2008*; *Rombaux et al., 2010*), we tested for such correlations in these data. The one olfactory test that was completed by all 46 scanned participants was the odorant identification test, which revealed no correlation with bulb volume (uRPL: n = 23, Spearman Rho = 0.25, p=0.25. Control: n = 23, Spearman Rho = 0.146, p=0.507. Combined: n = 46, Spearman Rho = 0.068, p=0.652). To further probe this issue, we assigned an olfaction composite score to each participant, reflecting all the olfactory tasks she completed (see Methods). Using this more comprehensive olfactory score, we observed a significant correlation in the uRPL cohort alone (uRPL: n = 22, Spearman Rho = 0.45, p=0.037; Control: n = 23, Spearman Rho = −0.055, p=0.8; Combined: n = 45, Spearman Rho = 0.065, p=0.67) (*Figure 4e*).

Next, in a hypothesis-free analysis, we conducted whole-brain voxel-based morphometry (VBM). This analysis, with correction for the multiple comparisons associated with the entire brain, uncovered no significant group differences. We note that if we repeat this analysis at strict threshold (p<0.0001) but without correction for multiple comparisons, it implied reduced gray matter in the right fusiform in uRPL (*Figure 5—figure supplement 1*). Hence, structural brain imaging uncovered a significant difference in the olfactory structures (olfactory bulb and olfactory sulci) of women who experience uRPL, and hinted at a possible difference in non-primary olfactory structures (right fusiform), yet that have been implicated in social chemosignaling (*Zhou and Chen, 2008*). We next turn to functional imaging.

## Women with uRPL have an altered brain response to men's body-odor

We used functional magnetic resonance imaging (fMRI) in 23 women with uRPL and 23 controls who watched varying-arousal movie-clips during subliminal exposure to body-odor from a *Non-Spouse men*. We chose not to use spouse body-odors in this particular experiment because this may have presented a confound: As noted, women with uRPL can recognize their spouse's body-odor. This pronounced difference between cohorts was overwhelmingly apparent in the previous rating experiment, where we can mention anecdotally, that although blind to condition, upon presentation with spouse body-odor, 15 of 33 women from the uRPL cohort spontaneously said 'oh, that is my spouse'.

This never happened once in the control cohort. Thus, if we used spouse odors in the MRI, we would risk the confound of only uRPL women knowing that any odors, let alone their spouse body-odors, were presented. Moreover, given that uRPL relationships may involve significant emotional strain (*Kolte et al., 2015*), this potential confound would be exacerbated. Thus, we chose to use male body-odor collected from donors unrelated to all study participants, and use the odor at subliminal levels that were not spontaneously detected. We further decided to test these body-odors as modulators of an emotional response rather than an activating stimulus alone because the impact of human social odors is typically more evident in this manner (*Jacob et al., 2001*).

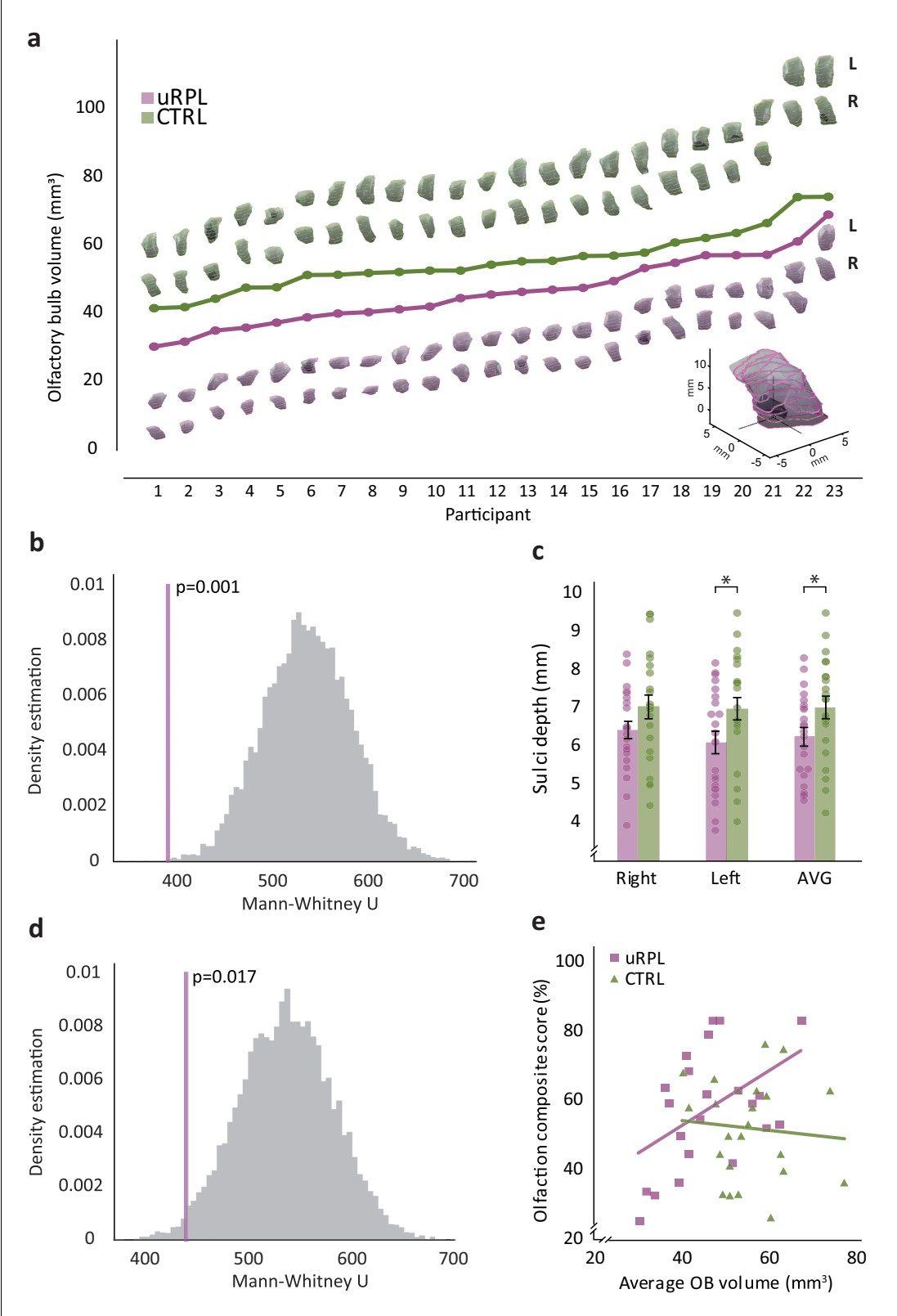

**Figure 4.** Women with uRPL have smaller olfactory bulbs and shallower olfactory sulci. (a) Olfactory bulb volume. A 3D reconstruction of uRPL (purple) and control (green) participants' left (upper row of the two) and right (bottom row) olfactory bulbs. Bulbs sorted by size (see *Figure 4—figure supplement 1* for sort by participant (uRPL to Control) match). Note that the reconstructions do not relate to the values on the Y axis, the values are reflected in the data lines alone. (b) Bootstrap test of 'a', 10,000 repetitions, the purple line represents the actual uRPL-control difference value. (c)

*Figure 4 continued on next page*

*Figure 4 continued*

Olfactory sulci depth (right, left and average) of uRPL (purple bars) and control (green bars) participants. (d) Bootstrap test of the average in 'c', 10,000 repetitions, the purple line represents the actual uRPL-control difference value. Bar graphs depict means, each dot represents a participant, dots are jittered to prevent overlay. Error bars are s.e.m. *p<0.05. (e) The relation between olfactory composite scores and olfactory bulb volumes for uRPL (purple) and control (green) participants. n = 46. Each square (uRPL) and triangle (control) represent a participant.

The online version of this article includes the following source data and figure supplement(s) for figure 4:

**Source data 1.** All olfactory bulb volumes and sulci depth.

**Figure supplement 1.** Smaller olfactory bulbs in uRPL.

The functional study was made of two consecutive scans, one with olfactometer-delivered undetected non-spouse men's body-odor, and one with olfactometer-delivered blank (counterbalanced for order across participants). In each such scan, each participant observed 40 12s-long movie clips, and after each, participants self-rated the level of emotional arousal the clip induced (using an 8-level response box). Like in the structural analysis, in the functional analysis we also applied both a targeted and hypothesis-free approach. In the targeted approach, we first concentrated on the hypothalamus, the primary brain structure implicated in linking the brain olfactory and reproductive systems (*Rosser et al., 1989*; *Yoon et al., 2005*; *Brennan, 2009*). The hypothalamus has also been indicated in women's responses to a molecule (AND) that may occur in men's body-odor (*Savic et al., 2001*). We used a hypothalamic region of interest (ROI) independently defined in Neurosynth (*Yarkoni et al., 2011*), and applied a contrast of Group (uRPL/Control) and Stimulus (Undetected body-odor/Blank) with parametric modulation for the self-rated level of emotional arousal. Following small-volume correction for multiple comparisons, we observed a remarkable effect in the hypothalamus (Z > 2.3, Cluster-corrected) (*Figure 5a*). Although the p-value associated with this result is the statistical parametric map statistic (p<0.01, corrected), we used a repeated-measures ANOVA applied to the parameter estimates only to understand the directional drivers of this effect. We observed that whereas body-odor increased hypothalamic activation in uRPL (t(22) = 2.81, p=0.01), it decreased hypothalamic activation in controls (t(22) = 2.41, p=0.025), and the interaction was powerful (F(1,44) = 13.4, p=0.0007) (*Figure 5a*). In turn, a hypothesis-free approach full-brain contrast ANOVA with correction for the multiple comparisons involved, did not uncover additional brain regions differently impacted by arousal as a function of exposure to body-odor in uRPL vs. controls.

Given the hypothalamic group-difference, we next used psychophysiological interaction (PPI) analysis (*Friston et al., 1997*) to ask whether functional connectivity with this region differed across groups as a function of exposure to body-odor. We observed that with increased arousal, body-odor increased connectivity between the hypothalamus and right insula significantly more in controls than in uRPL (peak Montreal Neurological Institute coordinates (MNI): [33,21,-8], 2336 voxels, Cluster-corrected z > 3.1) (*Figure 5b*) We again note that the p-value associated with this finding is the above PPI mapping statistic (p<0.001, corrected), but to only verify the directionality of the effect we applied a repeated-measures ANOVA on the parameter estimates, revealing that the decrease was significant in uRPL (t(22) = 2.45, p=0.022), the increase was significant in Controls (t(22) = 4.41, p=0.0002), and the interaction was powerful (F(1,44) = 20.6, p=0.00004) (*Figure 5b*). Patterns of brain activity can be impacted by anxiety (*Holzschneider and Mulert, 2011*), which can be increased in uRPL (*Mevorach-Zussman et al., 2012*). To address this possible source of variance in our data, all participants completed the trait-anxiety questionnaire (State-Trait Anxiety Inventory; STAI *Teichman and Melnick, 1979*), and a personality questionnaire (The 'Big Five' Inventory *Etzion and Laski, 1998*). We observed that both uRPL and controls had moderate anxiety levels, and there was no significant difference between the groups (STAI: uRPL = 39.5 ± 8, Control = 39.87 ± 10.86, SW = 0.97, p=0.3, t(44) = 0.14, p=0.89, Cohen's d = 0.041).

Moreover, we observed no significant differences in personality (Extroversion (E), Conscientiousness (C), Neuroticism (N) Agreeableness (A), Openness to experience (O)); uRPL: E = 3.67 ± 0.59, C = 3.71 ± 0.49, N = 3.05 ± 0.67, A = 3.87 ± 0.46, O = 3.58 ± 0.67. Control: E = 3.51 ± 0.64, C = 3.94 ± 0.41, N = 2.97 ± 0.65, A = 3.73 ± 0.64, O = 3.73 ± 0.44 (All SW >0.957, all p>0.092, all t (44) < 1.68, all p>0.1, all Cohen's d < 0.5). Therefore, the unique pattern of brain activity in response to body-odor in uRPL was not a reflection of differences in anxiety or personality across cohorts.

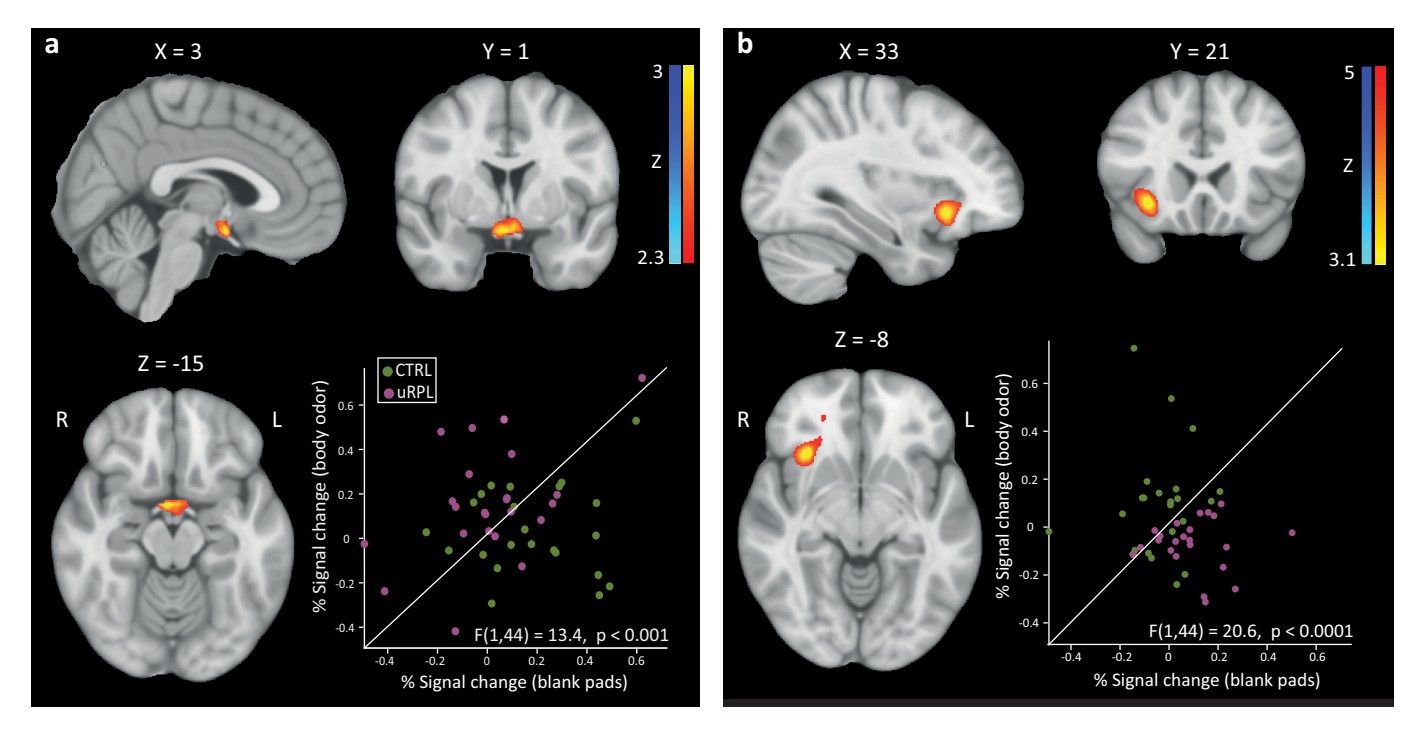

**Figure 5.** Women with uRPL have an altered brain response to male body-odor. (**a**) Hypothalamus blood-oxygen level-dependent (BOLD) activity in uRPL (non-spouse >blank), compared to control. (**b**) Whole brain PPI test, reflecting greater correlation with hypothalamus (seed ROI) time series for all emotionally-weighted-movie-clips>fixation. Both scatterplots reflect the % signal change of each participant (uRPL in purple, controls in green). The diagonal is the unit slope-line (x = y). Dots above the slop line represent higher % signal change for *Non-Spouse men* body odor, and dots below the line represent higher % signal change for *Blank*. n = 46. All coordinates in MNI space.

The online version of this article includes the following source data and figure supplement(s) for figure 5:

**Source data 1.** Parameter estimates for body-odor and blank.

**Figure supplement 1.** Reduced gray matter volume in the right fusiform in uRPL.

Finally, given these perceptual and brain differences, one may postulate a generalized different impact of body-odor in women with uRPL. To investigate this, we replicated an experiment where we previously observed an impact of body-odors on behavior and psychophysiology (*Endevelt-Shapira et al., 2018*). We tested 18 uRPL women and 18 controls in a single experimental session that included sniffing body-odors, followed by a face recognition task, an empathy task, and an emotional Stroop task, concurrent with Galvanic Skin Response (GSR) recording, once under exposure to Non-Spouse-men body-odor and again under exposure to Blank. Despite some potentially meaningful group and odorant differences, we failed to observe any significant interaction of group and odorant in this composite set of experiments. In other words, body-odors had the same impact on these measures in both uRPL and control women. Although one cannot read too much into such a null effect, it is important to report it when conveying our results, as it confines the extent of difference between these cohorts in their responses to body-odors.

## Discussion

We found that, as a group, women who experience uRPL have an overwhelming advantage over controls at recognizing their spouse by smell, yet only a slight general olfactory advantage. Whereas control performance at olfactory spouse identification was on par with two previous reports (*Hold and Schleidt, 1977*; *Mahmut et al., 2019*), it was poorer than in a third previous study, where participants performed on par with the current uRPL group (*Lundström and Jones-Gotman, 2009*). We speculate that these differences across studies reflect methodological differences impacting task

difficulty. More specifically, the latter study reapplied body-odor collection-pads to donors for seven straight nights. This likely made for a very concentrated body-odor. In contrast, our donors wore t-shirts for two nights, making for a less concentrated stimulus, and hence lower overall performance. The question here, however, was not of a comparison across studies, but rather across groups in the same (our) study, where we obviously applied common methods for both groups.

We note two considerations related to this primary behavioral finding. First, considering the only marginal uRPL advantage at other olfactory tasks, this advantage may reflect some downstream uRPL advantage, perhaps even abilities at memory beyond the olfactory system alone. Second, as mentioned in the results, this advantage may be impacted by a unique body-odor in uRPL men. Alas, our experimental design did not permit addressing this question in the current results, and this remains an open and tantalizing question for future research. That said, if there is a body-odor difference in uRPL men, we speculate that this is in addition and not instead of altered body-odor perception in uRPL women. We say this because in addition to better performance in spouse identification, uRPL women also reported different perceptual qualities associated with a group of independent men: Women with and without uRPL rated the same group of non-spouse men, and most women with uRPL perceived them differently (*Figure 3*). Thus, we stand by the notion that, at the group level, women with uRPL have altered perception of men's body-odor (*Figures 1c* and *3b*).

uRPL was also associated with significantly increased body-odor-induced activation in the hypothalamus, followed by altered patterns of functional connectivity, primarily with the insula. The hypothalamus may be thought of as the primary brain structure associated with reproductive behavior and state (*Schally et al., 1972*; *Maffucci and Gore, 2009*), and its link with the insula may be tied to processing human body-odors related to emotional state (*Prehn-Kristensen et al., 2009*), and in body-odor-based human kin recognition (*Lundström et al., 2009*). This combines with evidence of hypothalamic activation by androgen-like odorants in women but not men (*Savic et al., 2001*) to imply that body-odors may activate hormonal cascades in humans similar to those at the heart of several social chemosignaling effects characterized in rodents. This is consistent with the growing evidence for social chemosignaling in general human behavior (*McClintock et al., 2001*; *Prehn et al., 2006*; *Zhou and Chen, 2009*; *Mitro et al., 2012*; *Semin and Groot, 2013*; *Olsson et al., 2014*; *Lübke and Pause, 2015*; *de Groot et al., 2017*).

Although the functional results dovetail with the behavioral results to depict heightened processing of body-odors in uRPL, the structural results are counter-intuitive. uRPL was associated with smaller olfactory bulb volume. On one side, this result provides yet additional and particularly convincing evidence for an altered olfactory brain profile in uRPL. We label this as particularly convincing because volumetric differences are less susceptible to analysis strategies than are functional differences. On the other hand, whereas smaller OBs are typically associated with poorer olfaction (*Buschhüter et al., 2008*; *Rombaux et al., 2010*), here we observed smaller OBs in the group with better olfactory spouse identification (uRPL), and slightly better olfaction overall. A correlation between OB volume and general olfactory performance materialized only within the uRPL group, but not in the control group. We note that the relation between OB volume and olfactory performance is not always straight forward (*Weiss et al., 2019*), and we indeed have no explanation for this relationship in the current data.

We would like to also highlight several limitations of this study. First, the study with all its experiments was conducted over a very long period of time: nearly 7 years. This may have one advantage to it in that the results replicated at the hands of different experimenters over time, but the disadvantage is that methodological nuances were more varied across participants than one would want. This was an unnecessary source of variance. Second, some of the behavioral experiments were not double-blind in the strictest sense of the term. More specifically, although participants received instructions and entered replies through computer, the experimenter that placed the stimuli before them knew stimulus identity. A third limitation is the lack of correlation within participants across olfactory tasks. One typically expects participants who are good at one olfactory task to be good at other olfactory tasks, yet here these correlations were weak, a weakness we have no explanation for. Finally, the fMRI results were evident in targeted ROI-based analyses, but did not emerge from hypothesis-free strict parametric mapping. Although this is not unusual, especially with relation to hypothalamic results (*Savic et al., 2005*; *Burke et al., 2012*; *Burke et al., 2014*), it is less powerful than one would want. Despite these limitations, we think the overall image of an altered olfactory brain, and altered olfactory function in uRPL, particularly in relation to body-odor, holds strong.

Finally, it is tempting to combine the current picture of altered perception and brain responses to men's body odor in uRPL, with additional findings such as a specific polymorphism in an olfactory receptor gene (OR4C16G > A) that has been associated with uRPL (*Ryu et al., 2019*), and the tradition of protective isolation of women during early pregnancy evident in some tribal societies (*Van Gennep, 1960*), to jointly hint at a form of Bruce effect in humans. In the Bruce effect, an exposure to the odor of a non-sire male frequently leads to implantation failure in females (*Bruce, 1959*). Such an analogy, however, is restricted by major limitations. First, correlation is not causation, and altered behavioral and brain responses to body-odors in uRPL, convincing however they may be, do not alone imply that this is causal in the condition. Behavior may differ because of increased motivation in participants with uRPL, and brain structure and function may differ because of an independent covarying factor that we failed to identify. Although we could think of several experiments that would investigate causation, these are prevented by ethical considerations. Moreover, any notion of a human Bruce-like effect is further challenged by the underlying neuroanatomy. In rodents, the Bruce effect relies on two neural structures that humans may not possess: The first is the VNO, namely the dedicated sensory epithelia in the rodent nasal passage that is primarily involved in social chemosignaling (*Keverne, 1999*). Humans may retain a VNO pit on the nasal septum (*Trotier et al., 2000*), but this structure is considered vestigial in the human nose (*Meredith, 2001*). Second, the rodent VNO projects to the rodent accessory olfactory bulb (AOB), a structure also reported missing in the human brain (*Meisami et al., 1998*). It is in the AOB where the odor of the fathering male may be imprinted during mating in rodents (*Brennan, 2009*). Although humans may lack a VNO and AOB, humans do functionally express vomeronasal receptors in their olfactory epithelium (*Rodriguez et al., 2000*). Although rodent social chemosignaling is primarily attributed to the VNO and AOB, the main epithelium and bulb can also support social chemosignaling (*Spehr et al., 2006*). Therefore, the missing neural substrates do not altogether preclude a human Bruce-like effect, but do restrict such an analogy. Finally, whereas in the Bruce effect the pregnancy is terminated at the stage of implantation, in humans this would be almost impossible to identify, and women who are diagnosed with uRPL are past the implantation phase (*Zipple et al., 2019*). With these limitations in mind, we restrict our claims to the observation of altered perceptual and brain responses to men's body-odor in uRPL. This potentially provides for a much-needed reorientation of thinking in uRPL. Rather than looking at the uterus and the hormonal environment alone, we looked at the brain, and more specifically at the olfactory system, and identify unique patterns associated with uRPL. Indeed, as far as we know, structural and functional patterns of brain organization have not been previously associated with uRPL, and this discovery was guided by our investigation of olfaction. Thus, this framework and set of findings may combine with additional results on the potential impact of social environment on pregnancy loss (*Catalano et al., 2016*; *Catalano et al., 2018*) to jointly redirect thinking on this condition, which is currently not well-understood, or managed. With all that said, we will end in a sort of disclaimer, which is important in light of the subject matter: The relationships we identified were group level relationships. As a group, women with uRPL differed from controls, but at every given measure, there were women from the uRPL group with control-like results, and women from the control group with uRPL-like results. Given this variability and overlap, this manuscript does not contain results that merit any behavioral adjustments by women trying to maintain a pregnancy.

## Materials and methods

### Participants

Conducting studies in uRPL is complicated by the issue of long-term continuous participant availability. This manuscript contains many different experiments that ideally would have all been conducted in the same cohort. This would have allowed optimal relating of one result to another. However, ethical considerations prevent us from conducting any experiments with women who are currently pregnant, or trying to conceive. Because this cohort is often doing exactly that, it is almost impossible to conduct experiments in a given participant over an extended period of time. Instead, participants were available sporadically. To deal with this, we recruited an overall cohort of 40 women with uRPL (mean age: 35.06 ± 6.22), and 57 controls (mean age: 34.66 ± 4.24), and then assigned matched pairs based on uRPL availability per experiment (*Supplementary file 1*). We set the sample sizes

based on an initial power analysis based on a previously published study (*Lundström and Jones-Gotman, 2009*). This implied that at alpha = 0.05% and 80% power we need to test a sample of 29 participants per group. All participants provided written informed consent to procedures approved by the Tel- Hashomer (behavioral studies, protocol reference number 8067–10-SMC) or Wolfson hospital (imaging studies, protocol reference number 0121–10-WOMC) Helsinki Committees. All participants were screened for good general health and no history of psychiatric disease.

### Inclusion/exclusion criteria
Inclusion criteria for uRPL was at least two consecutive miscarriages prior to 15 weeks, with an unknown etiology for the pregnancy loss, despite careful medical investigation. Inclusion criteria for controls was no history of (known) pregnancy loss or abortions.

### Control matching
Because this study involves natural groups rather than randomization, we paid close attention to controlling for factors which may confound our findings. For each comparison performed, we matched control participants to uRPL participants based on their age and number of children (see *Figure 1—source data 1*, *Figure 2—source data 1* and *Figure 4—source data 1* for details). If multiple choices were available, our choice was based on matching the average of the control group as a whole. For the olfactory bulb analysis, following recent findings regarding the impact of pregnancy on maternal brain anatomy after birth (*Kim et al., 2010*; *Hoekzema et al., 2017*; *Kim et al., 2018*), we additionally matched for age of offspring, particularly age of the youngest child (see *Figure 4—source data 1*). Additional factors, like marital status and living arrangements, were similar between groups: the overwhelming majority of women in our study were married and living with their spouses. One factor that is by definition confounded with group differences is number of pregnancies, as RPL is defined as two or more consecutive miscarriages. To ask whether number of pregnancies may be associated with our results, we examined the most prominent behavioral result, namely spouse identification, and its association with number of pregnancies within each group. We observed no significant correlations (uRPL group: Spearman's Rho = 0.19, p=0.3; Control group: Spearman's Rho = −0.29, p=0.11), implying that number of pregnancies alone does not impact this result.

### Behavioral studies
#### Body-odor collection
Each spouse was provided with a new 100%-cotton white T-shirt for body-odor collection according to standard procedures (*Endevelt-Shapira et al., 2018*). Briefly, the donors were instructed to wear the shirt for two consecutive nights. The donors were further instructed to avoid eating ingredients that can alter body-odor (fenugreek, asparagus, curry, etc.) prior to body-odor sampling. In addition, during the sampling days, donors were asked not to use soap, shampoo, conditioner or deodorant. Between the two nights, the T-shirts were kept inside a closed glass jar at room temperature, and after the second night, they were further stored at −20 ˚C to prevent bacterial growth.

#### Shirt sniffing device (SSD)
On the morning of the experiment shirts were thawed at room temperature inside the jars to avoid humidity condensation. Using sterile scissors, shirts were cut into two longitudinal pieces, such that each half of the shirt contained one axillary area. Each half shirt was then placed inside an SSD – a glass jar covered by a cap with an air filter, inhalation mask and a one-way flap-valve (*Figure 1a*). As a blank control, new unworn shirts were also frozen inside a glass jar, thawed, and cut to two halves prior to the experiment.

#### SSD experimental design
At the onset of every trial, the experimenter arranged the three SSDs in front of the participant, and then moved away. One SSD contained a shirt that originated from the participant's spouse, one SSD contained a shirt that originated from another participant's spouse (non-spouse), and the third contained a Blank. The participant then received on-screen instructions to sniff each jar only once, before entering her selection of 'spouse' using a computer track-pad. Participants did not receive

feedback as to whether they were right or wrong. Each triangle combination was presented either four or six times (*Supplementary file 2*), and percent accuracy was calculated. Although this experimental arrangement is not genuinely double-blind (as the experimenter placing the jars knew their content), the lack of verbal interaction between the experimenter and participant during trials, the participant self-initiated sampling action and timing, and the use of different experimenters across participants, together minimize the risk of experimenter-generated cues.

## Odorant identification

We assessed ordinary odorant identification using a subset of an established and validated standardized test (University of Pennsylvania Smell Identification Test, UPSIT *Doty et al., 1984*). Although the full UPSIT contains 40 stimuli, the results with a subset of 20 stimuli are highly correlated to the results with 40 (*Doty et al., 1989*). To also verify this in our type of cohort, a subset of 26 uRPL participants completed the full 40 stimuli version. We observed a strong fit between the score with 20 and 40 odorants (Spearman Rho = 0.71, p<0.001). With this in hand, we proceeded to use the 20 stimuli test. Out of 97 participants (40 uRPL), three were excluded (2 uRPL) (>2.7 SD), retaining 94 participants (38 uRPL).

## Monomolecules

Each trial contained sequential presentation of three jars (counterbalanced for order), two containing the carrier alone (Propylene glycol) and one containing the carrier with the monomolecule: either androstadienone (AND, androsta-4,16,-dien-3-one, at a final concentration of 0.000025M), androstenone (ANN; 16, (5$\alpha$)-androsten-3-o, at a final concentration of 0.000025M) or estratetraenol (EST; 1, 3, 5 (10), 16-estratetraen-3-ol, at a final concentration of 0.045M). Participants were allowed to take one 2 s long sniff at each odorant presentation and were then asked to pick out the jar that contained the dissimilar odorant. A subset of participants (same number of uRPL and Controls) completed three repetitions per triangle, and the remainder completed 10 repetitions per triangle. The order of presentation of each triplet was randomized, and both the experimenter and participant were blind for condition.

## DMTS threshold test

Olfactory detection thresholds for the alliaceous odor dimethyl trisulfide (DMTS) were determined, using a maximum-likelihood adaptive staircase procedure (MLPEST) (*Linschoten et al., 2001*), in which a participant was presented with two jars, one blank (containing carrier alone; Propylene Glycol) and one containing a changing concentration of DMTS, and had to choose which contained the odorant (forced choice). Out of 39 participants (19 uRPL), two were excluded (1 uRPL, all >2.7 SD), retaining 37 participants(18 uRPL).

## Olfactory composite score

The composite score of identification, detection and threshold was calculated using the averaged monomolecule discrimination score, odor identification scores and threshold scores. The two latter scores were normalized to a scale of 0–1 (using the formula (x-min)/(max-min)), and all three scores were then averaged per participant to generate the comprehensive composite score. Out of 97 participants (40 uRPL), 1 uRPL was excluded (>2.7 SD) retaining 96 participants (39 uRPL). Next, we matched controls to uRPL women as described in the 'control matching' section, first by matching according to the type of tests these women have a score for, and if multiple options were available, by age and number of children. Two uRPL women did not have matched controls by test and age, so we matched controls by the closest type of tests performed and age, and removed the non-matching test from the composite average score of this participant.

## Body-odor ratings

The three body-odors, Spouse, Non-Spouse and Blank, were presented using SSDs to participants in randomized order. Participants were instructed to sniff for 2 s and then rate odor on a visual analog scale (VAS) for perceived intensity, pleasantness, sexual attraction and fertility associated with the body-odor. Inter-Stimulus Interval was 30 s.

## Behavioral statistics

All data analyses were performed using JASP (JASP Team (2019) Version 0.11.1). Each data-set was first tested for normality of distribution using Shapiro-Wilk test of normality. For normally distributed data, we used the following parametric measures: If only one variable was compared, we used two-tailed t-tests. To test for between-group differences across multiple related variables, we used multi-variate analysis of variance (ANOVA) followed by post-hoc two-tailed t-tests. To test for correlations between variables, we used the Pearson correlation coefficient for continuous variables, or Spearman's Rank Correlation Coefficient if one or both variables were categorical. For abnormally distributed data we used the following nonparametric measures: If only one variable was compared, we used Wilcoxon signed-rank test for one-sample and paired samples, and Mann-Whitney test for independent samples. To test for between-group differences across multiple related variables, we used linear mixed models (using R 3.6.1, packages readxl version 1.3.1, nlme version 3.1–140, car version 3.0–5) followed by the above-mentioned nonparametric tests, and tested for correlations using Spearman's Rank Correlation Coefficient. Finally, to estimate power, we calculated Cohen's $d$ for parametric measures and rank biserial correlation (RBC) for nonparametric measures. We note that in all cases where we relied on non-parametric tests, we also reported the parametric outcome in parenthesis for comparison. We observe that the non-parametric and parametric approaches yielded nearly identical outcomes in terms of assigning significance to the reported differences. Finally, we used bootstrap verification in order to verify the statistical validity of the more prominent results. In each case, we randomly shuffled the assignment of the data to uRPL or Control 10,000 times, and reconducted the analysis.

## Brain imaging

### Participants

A total of 48 participants (24 uRPL) were scanned in the fMRI study, and 55 participants (26 uRPL) in structural T2-weighted imaging for OB volume and OS depth analysis.

### Data acquisition

MRI Scanning was performed on a 3-Tesla MAGNETOM Trio Siemens scanner. For olfactory bulb (OB) volume and olfactory sulcus (OS) depth, a 32-channel head coil was used. Images were acquired using a T2-weighted turbo spin echo pulse sequence in the coronal plane, covering the anterior and middle segments. Sequence parameters: 35 slices, voxel size: 400 × 400 μm, slice thickness: 1.6 mm, no gap, TE = 85 ms, TR = 7000 ms, flip angle = 120°, two averages. Functional data were collected using a 12-channel head matrix coil and T2*-weighted gradient-echo planar imaging sequence, with the following parameters: 450 repetitions, TR = 2000 ms, TE = 25 ms, flip angle = 75°, FOV = 216 × 216 mm², matrix = 72 × 72 mm², voxel size = 3 × 3 mm, slice thickness = 3.7 mm, no gap, 34 transverse slices tilted to the AC-PC plane. Anatomical images for functional overlay were acquired using a 3D T1- weighted magnetization prepared rapid gradient echo (MP-RAGE) sequence at high resolution: 1 × 1×1 mm³ voxel size, TR = 2300 ms, TE = 2.98 ms, inversion time = 900 ms, and a flip angle = 9°.

All imaging data are publicly available at https://openneuro.org/datasets/ds002717.

### Structural brain imaging

Olfactory bulb and olfactory sulcus analysis: We excluded 3 of the 26 uRPL participants due to poor resolution (likely motion related) which did not allow a reliable measure of the sulcus or bulb. Next, 23 matching controls were selected based on parity and age of their children, specifically the age of their youngest child, following recent findings regarding the impact of pregnancy on maternal brain anatomy after birth (*Kim et al., 2010*; *Hoekzema et al., 2017*; *Kim et al., 2018*). We used independent sample t-tests to compare OB volume and OS depth between the two groups. OB volume and OS depth were measured according to standard methods (*Rombaux et al., 2010*). For delineating and measuring the OB we developed software that automatically identifies the region of the OBs, and then allows the user to manually delineate OB boundaries across interpolated images for automated volume estimation. This software was written by co-author LG, has been used before (*Weiss et al., 2019*), and is available for download at https://gitlab.com/liorg/OlfactoryBulbDelineation. To calculate estimated Total Intracranial Volume (eTIV) we used FreeSurfer version 6.0.0

(*Buckner et al., 2004*), next we normalize the OB by eTIV for 22 participants (MPRAGE for one uRPL participants was not obtained, so her matched control was also excluded).

OS depth was measured using ITK-SNAP version 3 (www.itksnap.org). The coronal slice was picked by the Plane of the Posterior Tangent through the eye-balls (PPTE), which in most individuals traverses the anterior-mid segment of the OB. In this slice, a virtual line that was tangent to the inferior border of the orbital and rectus gyri was drawn, and then perpendicular line connecting the above virtual line and the deepest part of the OS was marked. This line represents OS depth (*Rombaux et al., 2010*). Two independent raters, blind to participant identity and group-allocation, demarcated OB borders and marked OS depth. Next, all the OB volumes and OS depths in which between-rater difference was above 15%, were judged by a third blind review. Subsequent inter-rater correlation in OB volume was r = 0.92, p<0.001 and in OS depth r = 0.93,<0.001. Inter-rater agreement was further validated using intraclass correlation coefficient *Tinsley and Weiss, 1975*: OB volume: ICC r = 0.96, confidence intervals (CI) 0.925–0.981, OS depth: ICC r = 0.967, CI 0.939–0.985.

For voxel-based morphometry structural data from 23 uRPL and 23 matched-control was processed with FSL 6.0 software. First, the brain extracting tool (BET) was employed to cut the skull from all T1 MRI structural images *Smith, 2002*; Next, FSL Automated Segmentation Tool (FAST) was adopted to carry out tissue-type segmentation (*Zhang et al., 2001*). After careful check for segmentation quality, the segmented GM parietal volume images were aligned to the MNI standard space (MNI152) by applying linear image registration tool (FLIRT) and nonlinear registration (FNIRT) methods. A study-specific template was created by averaging the registered images, to which the native grey matter images were then non- linearly re-registered. The registered GM parietal volume images were modulated for the contraction/enlargement due to the nonlinear component of the transformation by dividing them by the Jacobian of the warp field. Next, the segmented and modulated images were smoothed with isotropic Gaussian kernels with a standard deviation of sigma = 3 mm. Finally, a voxelwise GLM was applied using permutation-based non-parametric testing, using p<0.0001, without additional correction for multiple comparisons.

## Functional brain imaging

### Odor stimuli

Twenty-two heterosexual men (mean age 31.1 ± 6.4) donated their body-odor, following the standard procedures described above, with two differences: we used eight adhesive pads rather than t-shirts, for four nights rather than 2. After each night the pads were stored at -20°C. We used pads rather than t-shirts so as to have smaller and more concentrated odorant sources that would fit into the olfactometer. These pads were cut into small pieces and mixed such that the scanner stimulus contained a mix of 20 men.

### Video clips

An independent group of 12 women rated the emotional arousal, on a scale of 1 to 10, of 30 ~ 1 min-long scenes. Scenes, all containing human characters, were chosen from 11 commercial films. From these we used the 20 scenes with the highest emotional scores. All scenes were edited into 12 s clips.

### Procedure

To assure no differences related to prior exposure to the film-clips, and to introduce the narrative and emotional context of the stimuli, prior to the scan, participants watched the 20 1-min-long scenes and rated familiarity. Next, a one-hour long fMRI scan was conducted, which included two functional scans, separated by a T1 anatomical scan. After the second functional scan, a T2 anatomical scan was conducted. Each functional scan started with a 4 s emotional clip (to be excluded in the analysis) and next 12-sec-long video clips were presented: 20 clips of high emotional content and 20 landscape clips, in alternating order, with an ITI (fixation point) of 8–12 s. Following each video clip, participants were asked to rate their emotional arousal, on a scale of 1 to 8, with eight being the highest. Simultaneously with each video clip, body-odor or blank were delivered using a computer-controlled air-dilution olfactometer that embedded the odorant pulse within a constant stream of clean air at 1.5 L/min. The two functional scans were identical except for the odor content. The two

conditions (blank vs. non-spouse-odor) were counterbalanced for order between participants. Participants were aware of the possibility of an odor being delivered but were not told when. After scanning, participants were asked to report whether there was an odor in each of the sessions, score of '1'=an odor (even if it was noticed only once) and '0'=no odor. First, we calculated the differences between body-odor sessions and blank sessions to receive three values: 0 (probability: 0.5),–1 (probability: 0.25), 1 (probability: 0.25). Next, we used a binomial test to calculate the probability of detection.

Following scanning, participants watched the emotional 12 s clips again, outside the scanner, and rated them for specific emotions: compassion, happiness, fearfulness, sadness, stress and emotional power, and familiarity prior to the experiment. There were no significant differences between the two groups in any of these ratings (all $t(44) < 1.39$, all $p>0.17$).

## Functional MRI analysis

Preprocessing and functional data analysis were conducted within FSL (FMRIB's Software Library, www.fmrib.ox.ac.uk/fsl), FEAT (FMRI Expert Analysis Tool) Version 6.00 (*Woolrich et al., 2001*), and MATLAB R2018a (MathWorks, Inc).

## Preprocessing pipeline

Two participants were excluded from the analysis: one uRPL woman due misunderstanding of the task (she did not rate the emotional arousal of the landscape clips), and one control participant due to miscarriage that was reported only after the scanning. Thus, a total of 46 participants, 23 in each group were submitted to further analysis. The first eight volumes (4 s clip) in each run were discarded to allow the MR signal to reach steady-state equilibrium. The structural brain images were skull-stripped using the FSL brain extraction tool (*Smith, 2002*). Functional images were corrected for slice-timing and head motion using six rigid-body transformations with FSL MCFLIRT (*Jenkinson et al., 2002*). Within each run, functional images were spatially normalized to the individual's anatomy and co-registered to the MNI 152 T1 template, using a combination of affine and non-linear registrations. Images were spatially smoothed with an 8 mm Gaussian kernel, and a high-pass filter (cut off = 128 s) was incorporated into the GLM to correct for scanner drift.

## Voxel-wise analyses

On the first level, all 3 EV regressors were modeled by a stick function convolved with a double gamma function. The regressors included: All video-clips with no weightings (and their temporal derivative), video-clips weighted with emotional arousal ratings (demean), reaction time (orthogonal to the other two regressors). This model also included a regressor of no interest for each volume, with >0.9 mm framewise displacement (using FSL motion outliers). At the second level, the parametric estimation of the weighted-clips was a contrast of non-spouse body-odor >blank using fixed-effects. Finally, for between group analysis we used small-volume correction of the hypothalamus (defined by Neurosynth), and FLAME1 (FMRIB's Local Analysis of Mixed Effects), with cluster-size correction z > 2.3 (p<0.01).

## Functional connectivity

We used psychophysiological interaction (PPI) analysis (*Friston et al., 1997*) to measure changes in functional connectivity modulated by emotional arousal. We conducted whole-brain PPI tests, reflecting greater correlation with hypothalamic (seed ROI) time series (physiological regressor) for all emotionally-weighted-movie-clips>fixation (psychological regressor). At the second level, the parametric estimation of the weighted-clips was a contrast of body-odor >blank using fixed-effects. Finally, for between group analysis, we used FLAME1, with cluster-size correction z > 3.1 (p<0.001).

## Acknowledgements

This study was funded by a European Research Council AdG. grant #670798 (SocioSmell) awarded to NS. General work in the Sobel lab is funded by the Rob and Cheryl McEwen Fund for Brain Research. Dr. E Furman-Haran holds the Calin and Elaine Rovinescu Research Fellow Chair for Brain Research. We thank Dr. Noa Ofen for fMRI consultation.

## Additional information

### Funding

| Funder | Grant reference number | Author |
| --- | --- | --- |
| H2020 European Research Council | 670798 | Noam Sobel |

The funders had no role in study design, data collection and interpretation, or the decision to submit the work for publication.

### Author contributions

Liron Rozenkrantz, Conceptualization, Data curation, Formal analysis, Methodology, Writing - original draft, Project administration, Writing - review and editing; Reut Weissgross, Tali Weiss, Data curation, Formal analysis, Investigation, Methodology, Project administration, Writing - review and editing; Inbal Ravreby, Formal analysis, Investigation, Writing - review and editing; Idan Frumin, Netta Reshef, Yael Holzman, Liron Pinchover, Data curation, Writing - review and editing; Sagit Shushan, Data curation, Formal analysis, Investigation, Writing - review and editing; Lior Gorodisky, Software, Formal analysis, Writing - review and editing; Yaara Endevelt-Shapira, Edna Furman-Haran, Data curation, Formal analysis, Writing - review and editing; Eva Mishor, Formal analysis, Writing - review and editing; Timna Soroka, Maya Finkel, Liav Tagania, Investigation; Aharon Ravia, Validation; Ofer Perl, Visualization; Howard Carp, Conceptualization, Resources, Data curation, Supervision, Project administration, Writing - review and editing; Noam Sobel, Conceptualization, Resources, Formal analysis, Supervision, Funding acquisition, Writing - original draft, Project administration, Writing - review and editing

### Author ORCIDs

Liron Rozenkrantz http://orcid.org/0000-0002-6496-4592
Reut Weissgross http://orcid.org/0000-0002-6554-9462
Idan Frumin http://orcid.org/0000-0002-6293-1586
Yaara Endevelt-Shapira https://orcid.org/0000-0002-9235-4572
Ofer Perl http://orcid.org/0000-0002-3560-4344
Noam Sobel https://orcid.org/0000-0002-3232-9391

### Ethics

Human subjects: All participants provided written informed consent to procedures approved by the Tel- Hashomer (behavioral studies, protocol reference number 8067-10-SMC) or Wolfson hospital (imaging studies, protocol reference number 0121-10-WOMC) Helsinki Committees. All participants were screened for good general health and no history of psychiatric disease.

### Decision letter and Author response

Decision letter https://doi.org/10.7554/eLife.55305.sa1
Author response https://doi.org/10.7554/eLife.55305.sa2

## Additional files

### Supplementary files

• Supplementary file 1. Participants. A table of all uRPL and Control women who participated in the project, including their age, number of pregnancies, number of repeated pregnancy losses (if any) and number of living children.

• Supplementary file 2. Spouse identification results. An extended table for '*Figure 1—source data 1*' containing the success rates at the spouse identification task, with the first two trials, first four trials, and all trials per participant.

• Transparent reporting form

## Data availability

Source data files have been uploaded for Figures 1-5. All imaging data is publicly available on Open-Neuro at https://openneuro.org/datasets/ds002717.

The following dataset was generated:

| Author(s) | Year | Dataset title | Dataset URL | Database and Identifier |
|---|---|---|---|---|
| Rozenkrantz L, Weissgross R, Weiss T, Ravrebi I, Frumin I, Shushan S, Gorodisky L, Reshef N, Holzman Y, Pinchover L, Endevelt-Shapira Y, Mishor E, Soroka T, Finkel M, Tagania L, Ravia A, Perl O, Furman-Haran E, Howard C, Sobel N | 2020 | Unexplained Repeated Pregnancy Loss is Associated with Altered Perceptual and Brain Responses to Men's Body-Odor | https://openneuro.org/datasets/ds002717 | OpenNeuro, ds002717 |

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
