## [Decision Letter]

**Acceptance summary:**

Human pregnancies often end in miscarriages, with most pregnancy losses remaining unexplained. Olfaction plays a key role in mammalian reproduction in general. Strikingly, Rosenkrantz, Sobel and colleagues now find that – as a population – women that experience unexplained repeated pregnancy loss have altered olfactory perceptual ability, reduced olfactory bulb volume, and increased hypothalamic response to men's body odours. This elegant study combines a variety of approaches to convincingly correlate olfaction with repeated pregnancy loss in humans, potentially providing a new entry point for clinical investigations. Furthermore, it highlights the importance and behavioural relevance of the sense of smell in humans in general.

**Decision letter after peer review:**

Thank you for submitting your article "Unexplained Repeated Pregnancy Loss is Associated with Altered Perceptual and Brain Responses to Men's Body-Odor" for consideration by *eLife*. Your article has been reviewed by five peer reviewers, and the evaluation has been overseen by a Reviewing Editor and Catherine Dulac as the Senior Editor The following individual involved in review of your submission has agreed to reveal their identity: Matthew N Zipple (Reviewer #4).

The reviewers have discussed the reviews with one another and the Reviewing Editor has drafted this decision to help you prepare a revised submission.

All reviewers agree that the data you present are both of high quality and importance. You present the intriguing finding that human females with repeated pregnancy loss (RPL) are better able to identify their spouse odour compared to controls. Moreover, you demonstrate differences in the anatomy of their olfactory system and differences in functional activity when exposed to male odours. All reviewers agree that the data – while correlative as you also discuss – are very clear, clearly presented, and compelling. However, there are major concerns that were consistently raised by virtually all reviewers (coming from very different fields):

Most importantly, while the correlations between olfactory function and RPL are compelling, the parallels to the Bruce-effect are not. Please do substantially tune down the Bruce effect analogies by restricting them to a discussion point and removing them from Abstract and Introduction. In the Discussion, however, it is necessary to have a more general discussion of the human (and in general primate) Bruce effect, its definition and the wider literature as outlined in particular by reviewers 4 and 5 (for that purpose I append the full reviews below).

Other major concerns (that you can all address with rewriting, adding raw data tables, and additional discussion):

• The Materials and methods indicate that for the experiments presented in Figures 1-2 each woman sniffed the SSD either 4-6 times. It is not clear why some women underwent two additional attempts. The “individual” data points represented in the figures are an average of these 4-6 trials. Please include a table of the raw data points for each individual to reveal the variance in response and identify the conditions where the additional trials occurred.

• Was the identity of the odor (spouse or stranger) blind to the researcher that presented the SSDs to each individual (I cannot find this in the Materials and methods)? If not, subliminal cues by the experimenter could alter the individual's perception and should be stated as a potential reason for behavioral differences observed.

• It is possible that the effects for spouse-odor identification are not at all related to RPL women's olfactory system but that they reside in their partners. As acknowledged by the authors, the men in RPL relationships may have unique body odors that are easy to be identified. One way to rule this out is to use the same set of odors in both groups, such that the spouse odor in one group is the non-spouse odor in the other. If the effects are driven by distinctive spouse-odors in the RPL group, women in both groups should be able to correctly identify this odor resulting in similar performance across groups. That is, the RPL group could correctly identify their spouse in the RPL group, and the control group could correctly identify these stand-out odors as the non-spouse odor. Ideally, the authors would rule out this important possibility in a control experiment. At the very least, however, this should be acknowledged and discussed. This confound is absent in many of the other experiments including the imaging studies. So while this potential explanation of the spouse-odor identification results needs to be taken seriously, it does not distract from the overall importance of this study.

• Because this study involves natural groups rather than randomization, there are a number of important confounds that need to be controlled for. For instance, groups may differ in the number of total pregnancies and other factors that are related to RPL. These factory could have an effect on olfaction in RPL women or on their ability to recognize their spouse's odor. Moreover, other factors may include the number of sexual partners, their living arrangements, etc. These and other factors could explain the present association and this should be tested as possible. In general, more needs to be said in the main text about how the groups were matched and how potential confounds were ruled out.

• A curious finding is the negative correlation between the composite olfactory score and the spouse identification performance in the RPL group. Would we not expect a positive correlation and importantly present across all participants?

Reviewer #1:

This manuscript describes a series of experiments testing for "Bruce effect"-like effects in humans. The key findings are that compared to controls, human females with repeated pregnancy loss (RPL) were able to identify the body odor of their spouses, despite relatively comparable olfactory perceptual ability. Compared to controls, these women rated stranger's body odors differently and also showed increased activity in the hypothalamus to such odors. In addition, women with RPL had significantly smaller olfactory bulbs.

This is a expertly conducted study on an important topic with interesting results. The findings are important because they may lead to new interventions for RPL. That said, there are a few issues that should be considered.

1) The effects for the spouse-odor identification test are strong and convincing. However, the results of the follow-up tests, aimed at identifying the specific cause of this advantage, are much weaker and less conclusive. From this perspective, it is less clear that women with RPL have an heightened sense of smell, and thus these findings should be interpreted with more caution. If there are no clear differences between groups, this opens up the possibility of alternative explanations. For instance, it is tempting to speculate that non-peripheral mechanisms, likely downstream of olfactory bulb are at play. Such effects could for instance be related to improved odor memory.

2) Alternatively, it is possible that the effects for spouse-odor identification are not at all related to RPL women's olfactory system but that they reside in their partners. As acknowledged by the authors, the men in RPL relationships may have unique body odors that are easy to be identified. One way to rule this out is to use the same set of odors in both groups, such that the spouse odor in one group is the non-spouse odor in the other. If the effects are driven by distinctive spouse-odors in the RPL group, women in both groups should be able to correctly identify this odor resulting in similar performance across groups. That is, the RPL group could correctly identify their spouse in the RPL group, and the control group could correctly identify these stand-out odors as the non-spouse odor. Ideally, the authors would rule out this important possibility in a control experiment. At the very least, however, this should be acknowledged and discussed. This confound is absent in many of the other experiments including the imaging studies. So while this potential explanation of the spouse-odor identification results needs to be taken seriously, it does not distract from the overall importance of this study.

3) Because this study involves natural groups rather than randomization, there are a number of important confounds that need to be controlled for. For instance, groups may differ in the number of total pregnancies and other factors that are related to RPL. These factory could have an effect on olfaction in RPL women or on their ability to recognize their spouse's odor. Moreover, other factors may include the number of sexual partners, their living arrangements, etc. These and other factors could explain the present association and this should be tested as possible. In general, more needs to be said in the main text about how the groups were matched and how potential confounds were ruled out.

4) A curious finding is the negative correlation between the composite olfactory score and the spouse identification performance in the RPL group. Would we not expect a positive correlation and importantly present across all participants?

Reviewer #2:

Rozenkrantz et al. investigate links between repeated pregnancy loss (RPL) and women's perception of male body odor, hypothesizing an underlying Bruce-like effect. The authors find several perceptual and neuroanatomical differences between RPL and control groups. These effects are certainly intriguing, especially the reported higher ability of RPL women to identify their spouses, as well as the neuroanatomical differences between this group and controls. However, these data do currently not allow to conclude a Bruce-like effect in humans. In my opinion, this study would therefore only be publishable in *eLife* conditional on addressing the concerns outlined below.

1) Extensive parallels are drawn to the Bruce effect, the mechanisms of which have most extensively been described in rodents. In mice, pheromone detection via the vomeronasal pathway and subsequent hypothalamic activation and dopamine release results in a loss of pregnancy only prior to implantation. In contrast, vomeronasal signalling is likely absent in humans and miscarriages (or pregnancies for that matter) would only be diagnosed long after embryo implantation. Even though the authors explicitly state that the Bruce effect is unlikely to be present in humans, they extensively refer to this phenomenon and e.g. place the hypothalamic activation in RPL women in this context. Although this limited equivalence is being addressed in the Discussion, allusions to a Bruce(-like) effect should feature less prominently throughout the Abstract/Introduction and Results sections.

2) The authors' use targeted analysis of their structural and functional imaging data to identify differences (OB volume, hypothalamic activation) between RPL women and controls. Yet, when the same data are analyzed in a hypothesis-free manner, the same differences cannot be identified (apart from when data are not corrected for multiple comparisons). This should be discussed.

3) A Bruce-like effect would rely on the correct identification of a smell as belonging to a Stranger. The study design chosen here however asked participants to distinguish between Spouse, Stranger and blank. While the authors argue that the correct identification of a familiar Spouse smell is a prerequisite for identifying other smells as Stranger, I am curious why participants were not asked to choose between e.g. two (or several) Spouse jars and one Stranger jar.

Reviewer #3:

In "Unexplained Repeated Pregnancy Loss is Associated with Altered Perceptual and Brain Responses to Men's Body-Odor" Rozenkrantz et al. explore the hypothesis that repeated pregnancy loss (RPL) in humans may be olfactory driven, similar to the Bruce effect in mice. When assayed for the ability of women to guess the odor of their spouse verses another male, they find the population of women that experienced RPL to perform better than women that were easily able to conceive. The correlation is extended to anatomical differences in the size of RPL olfactory bulbs and functional activity differences to male odors assayed by fMRI. Overall, the data is correlative, as is expected with human studies, there is no direct evidence that olfaction is contributing to RPL. As I read the manuscript, I tallied a list of major concerns, and then found them largely clearly stated in the last two paragraphs of the Discussion. The Bruce effect is only superficially analogous to the observations and experiments here, and the claims are much more limited than what is implied in the Abstract. With such a devastating condition as RPL, I do not think the story should be speculatively comingled with a unique aspect of behavior of a subset of species. I recommend removing most references to the Bruce effect prior to publication.

1) Though there is a statistical difference between RPL and control in Figures 1A/2/3B the response of both populations are very broad and the overall spread is very similar between RPL and control women. This looks like an example of a coincidental statistical difference – but not a biologically meaningful difference. I expect that if one was to look across all women (impossible) the results would not hold. With such a large variance in both populations in all measures of study, I do not think the data supports the conclusions.

2) The Materials and methods indicate that for the experiments presented in Figures 1-2 each woman sniffed the SSD either 4-6 times. It is not clear why some women underwent two additional attempts. The “individual” data points represented in the figures are an average of these 4-6 trials. Please include a table of the raw data points for each individual to reveal the variance in response and identify the conditions where the additional trials occurred.

3) Was the identity of the odor (spouse or stranger) blind to the researcher that presented the SSDs to each individual (I cannot find this in the Materials and methods)? If not, subliminal cues by the experimenter could alter the individual's perception and should be stated as a potential reason for behavioral differences observed.

4) The functional fMRI differences are confounded by watching video clips during assay. Why is this necessary? Of relevance here is the activity response towards odors only. The conclusions would be stronger if the stimuli were clear and controlled.

Reviewer #4:

The data that you present are of high quality and may be of great value to the RPL/medical literature, although I am unable to assess that aspect of your paper in detail. My reviewing expertise lies in the connection between your manuscript and the Bruce effect literature, and I believe that you would be better off avoiding discussion of the Bruce effect entirely. It is unclear that there is or should be any mechanistic or functional connection between RPL in humans and the Bruce effect, and I fear the addition of this manuscript to the Bruce effect literature would do more to confuse than to clarify.

I don't wish to belabour the below points more than I already have. In areas where my explanation may lack detail, I would recommend a thorough read of

deCatanzaro (2015) "Sex steroids as pheromones in mammals: the exceptional role of estradiol." Hormones and Behavior.

and

Zipple et al. (2019) Male-mediated prenatal loss: Functions and mechanisms." Evolutionary Anthropology.

1) The mechanistic argument: In order for use of the term "Bruce effect" or even "Bruce-like effect" to be justified, it seems necessary to show that there is some reason to believe that terminations of pregnancies are occurring after women are exposed to non-sire males.

You attempt to make this connection by showing that, compared to control women, women that display RPL are (1) more discriminating in their olfaction, especially their ability to distinguish between a spouse and a non-spouse, (2) have smaller olfactory bulbs, and (3) show differential brain response to men's body odors. The underlying logic appears to be that women who display RPL are sensitive to the smell of other men, and after smelling a non-spouse man, a chemical cascade is triggered that results in the termination of their pregnancies.

Such a line of argumentation would be much firmer if we had reason to believe that exposure to the odors of non-sire males is the trigger that leads to pregnancy termination across the many species that exhibit the Bruce effect, including some primates. Under such a scenario the lack of causality in the paper might justify the speculation that the presented data support a Bruce effect in humans. However, at present there is no evidence in any species that pregnancy termination (as opposed to implantation failure) is triggered by exposure to the scents of non-sire males. Even in mice, where the odor of non-sire males has been argued to trigger a Bruce effect by preventing implantation, it is unclear that exposure to the odor alone is sufficient to trigger the Bruce effect in the absence of exogenous estradiol.

What's more, it seems unlikely that such a complex and costly behavior of pregnancy termination could be triggered in humans simply as a result of a conspecific chemical cue, especially in a species in which females routinely encounter non-spouse males. Outside of rodents, there has so far been no evidence that odor or any other exogenous chemical mechanism is at play in species that exhibit the Bruce effect, such as dogs, horses, or primates.

While it is possible that olfactory cues play a role in post-implantation Bruce effects, it is also possible that the relevant cues in these cases are entirely social/visual and require no chemical cue whatsoever. Animals that are capable of recognizing each other visually, both because they have the cognitive capacity for such an assessment and because they live in a visual environment that allows for it, seem much more likely to rely on sociocognitive rather than exogenous chemical cues to trigger termination. A woman (or for that matter a female horse, dog, or lion) does not need to smell a non-sire male to know that he is present in her social group nor does she need to smell a male to know that he might pose a threat to the survival of her future children. Rather, she is able to assess changes in the dynamics of her group to make such determinations in a much more accurate way.

So, from a mechanistic standpoint, the speculation about associations between olfactory sensitivity and RPL does not appear justified.

2) The evolutionary argument: I also have concerns about the evolutionary story that you are telling here. Your argument appears to be that the ability of control women not to miscarry following exposure to the scent of non-sire men is an adaptation to communal living. The implication appears to be that termination following exposure to the scent of non-sire males is the ancestral state in primates, a claim that seems unlikely given the above discussion of mechanisms. It is also a confusing claim, because a great many primates live "communally," but there are certainly specie in which the Bruce effect does not occur (e.g. chacma baboons and yellow baboons). So when did this adaptation occur? Was it a prerequisite for group-living? If so, then it must have occurred millions of years ago.

If we take this interpretation as accurate, the unstated evolutionary argument that you are making is that the ancestral chemical cascade that had previously induced termination following exposure to non-sire males was broken via a mutation in an olfactory receptor gene (OR4C16G>A). For millions of years since that cascade was broken, it has remained otherwise entirely functional except that this change in one olfactory reception gene has caused it to no longer be triggered. So, when women are born with a mutation in the olfactory reception gene that makes it functional once more, the necessary chemical pathways that lead from exposure to male odors to pregnancy termination kick in and continue to work just as they would have millions of years previously. This seems to me to be a very unlikely scenario. There are additional issues that I see in terms of the evolutionary narrative at play, but I will not belabor these points further.

I therefore see no reason to refer to the Bruce effect in this manuscript, except as perhaps a passing comment in the Discussion that suggests it is possible that there may be some connection. Even this feels like more speculation than may be warranted.

Reviewer #5:

This manuscript describes experiments suggested by the argument that human females exhibit a Bruce Effect in which pregnant women spontaneously abort gestation when exposed to the odor of men other those who impregnated them. The experiments involve exposing women with and without histories of repeated pregnancy loss to odors of partners as well as other men to determine, in general, whether those with such histories detect partner odor better than controls, whether their olfactory biology differs from controls, and whether they react differently to arousing stimuli when exposed to male body odors. The pattern of results support the argument that women with a history of repeated pregnancy loss appear different from controls in way consistent with a Bruce Effect in women.

Those of us inclined to believe that humans exhibit Bruce Effect like behavior will find your arguments and findings powerful. Skeptics will offer caveats many of which you anticipated, and both directly and indirectly defend against. But to further avoid capitalizing on confirmation bias among readers, I suggest you alert them to two circumstances. First, the literature that explains, rather than describes, the "Bruce Effect" characterizes the termination of gestation when the environment threatens the survival of offspring as adaptive. Natural selection presumably conserved mutations that produce the Effect because they increase the reproductive fitness of females. They should, therefore, appear frequently among females in a stable, self-sustaining population. You, however, study a clinically defined sub-group of the population. A sub-group that exhibits a non-adaptive behavior, very frequent pregnancy loss. Your findings could be characterized as identifying risk factors for inclusion in a relatively small group that does not contribute to a self-sustaining population (i.e., 2-4% of all women who suffer frequent clinically recognized fetal loss), rather than providing evidence for an adaptive mechanism that, if the Bruce Effect argument is correct, should appear in a large fraction of women. You need, in short, to discuss the external validity of your findings and its implications for theory.

Second, you should spend a bit of page space making clear that the literature includes tests of a generalized Bruce Effect in humans. That work shows that, consistent with the theory underlying the Bruce Effect, populations witnessing unexpected death among children suffer greater than expected male fetal loss. The work remains controversial because it invokes "Bruce Effect" to describe the termination of gestation following something other than exposure to the odor of men. But the work does allow you to argue that your reasoning and findings may apply to circumstances other than rare cases of repeated pregnancy loss.

---

## [Author Response]

Most importantly, while the correlations between olfactory function and RPL are compelling, the parallels to the Bruce-effect are not. Please do substantially tune down the Bruce effect analogies by restricting them to a discussion point and removing them from Abstract and Introduction. In the Discussion, however, it is necessary to have a more general discussion of the human (and in general primate) Bruce effect, its definition and the wider literature as outlined in particular by reviewers 4 and 5 (for that purpose I append the full reviews below).

We have removed the Bruce effect from the Abstract, Introduction, and Results, and now only carefully allude to it in the final paragraph of the Discussion. This was initially hard to do… because the Bruce-effect frame was not an after-thought of our results, but rather genuinely the impetus for conducting this study in the first place. It was our hypothesis. Thus, emotionally, it was hard to let go. However, now in reading our final revised version, I must admit that this was the right thing to do, and this made for a better manuscript.

Other major concerns (that you can all address with rewriting, adding raw data tables, and additional discussion):• The Materials and methods indicate that for the experiments presented in Figures 1-2 each woman sniffed the SSD either 4-6 times. It is not clear why some women underwent two additional attempts. The “individual” data points represented in the figures are an average of these 4-6 trials. Please include a table of the raw data points for each individual to reveal the variance in response and identify the conditions where the additional trials occurred.

Some participants had different numbers of trials for the following reason: This study was conducted over many years, and naturally, we evolved some of the paradigms on the way, in order to improve them. In this case, it dawned on us that it would be more powerful to be able to assess significance also within each participant alone and not only across groups, so we added trials per participant. That said, critically, the number of trials remained equal across cohorts. In other words, for every RPL participant with a reduced number or trials there was a matched control with equally reduced trials. Thus, this shortcoming cannot underlie the group effect that we report. Moreover, in consideration of this comment, we now tested again, using only the first four trials from all those participants that had 6 trials, and we obtain ostensibly the same results. Finally, there were 3 cases where we had only two trials (I will remind that these are 3AFC trials, not two alternative). If we restrict the entire cohort to just two trials, the effect again still remains significant. To conclude, the group effect we reported was not influenced by the number of trials conducted per participant. Finally, consistent with the reviewer suggestion, we have now included a table (Supplementary file 2) which details all trials for each participant, and we added these sub-analyses using restricted portions of the data for verification.

• Was the identity of the odor (spouse or stranger) blind to the researcher that presented the SSDs to each individual (I cannot find this in the Materials and methods)? If not, subliminal cues by the experimenter could alter the individual's perception and should be stated as a potential reason for behavioral differences observed.

This experimental design was not double-blind in the strictest sense of the term, but its arrangement largely prevented the risk of experimenter-generated cues. More specifically, during each trial the experimenter did not hand the participant one SSD at a time, but rather placed three arbitrarily marked SSDs on the table in front of the participant at the beginning of the trial. The experimenter then moved away, and the participant alone interacted with a computer that provided sampling instructions. The participant then entered their selection using the trackpad of the computer, such that the experimenter did not know what the participant answered at that time. There was no voice interaction between the experimenter and participant during a trial. In other words, the timing of each sniff, its content, and answer, were all independently controlled by the participant without experimenter involvement. Finally, we used different experimenters across participants. Nevertheless, we acknowledge that this is not genuinely double-blind, and moreover, we acknowledge that this is a drawback. Indeed, in retrospect, an unnecessary drawback. This has now been better detailed, and explicitly acknowledged, in the manuscript.

• It is possible that the effects for spouse-odor identification are not at all related to RPL women's olfactory system but that they reside in their partners. As acknowledged by the authors, the men in RPL relationships may have unique body odors that are easy to be identified. One way to rule this out is to use the same set of odors in both groups, such that the spouse odor in one group is the non-spouse odor in the other. If the effects are driven by distinctive spouse-odors in the RPL group, women in both groups should be able to correctly identify this odor resulting in similar performance across groups. That is, the RPL group could correctly identify their spouse in the RPL group, and the control group could correctly identify these stand-out odors as the non-spouse odor. Ideally, the authors would rule out this important possibility in a control experiment. At the very least, however, this should be acknowledged and discussed. This confound is absent in many of the other experiments including the imaging studies. So while this potential explanation of the spouse-odor identification results needs to be taken seriously, it does not distract from the overall importance of this study.

This is a keen observation, that we had indeed acknowledged in the manuscript, but apparently we had not done so sufficiently. That men in RPL relationships have a unique body-odor fingerprint is a tantalizing possibility. Alas, our experiments were not designed to address this question. Although the stranger in each experiment was indeed the spouse of a different participant (as here suggested), whether this stranger was an RPL-man or a control-man was not well balanced across groups (as we had not imagined this would be a factor). We are certain to conduct a future study probing this exact question, and here currently further stress this possibility as here advised. In addition to the note in the body of the manuscript, we now note in the Discussion:

“We would like to reiterate the possibility mentioned in the results whereby theoretically, this advantage may reflect not better olfactory memory/identification in uRPL women, but rather a unique body-odor in uRPL men. Alas, our experimental design did not permit addressing this question in the current results, and this remains an open and tantalizing question for future research. That said, if there is a body-odor difference in uRPL men, we speculate that this is in addition and not instead of altered body-odor perception in uRPL women. We say this because that in addition to better performance in spouse identification, uRPL women also reported different perceptual qualities associated with a group of independent men. Women with and without uRPL rated the same group of men, and the women with RPL perceived them differently (Figure 2). In combination, we conclude that women with uRPL have altered perception of men's body-odor.”

• Because this study involves natural groups rather than randomization, there are a number of important confounds that need to be controlled for. For instance, groups may differ in the number of total pregnancies and other factors that are related to RPL. These factory could have an effect on olfaction in RPL women or on their ability to recognize their spouse's odor. Moreover, other factors may include the number of sexual partners, their living arrangements, etc. These and other factors could explain the present association and this should be tested as possible. In general, more needs to be said in the main text about how the groups were matched and how potential confounds were ruled out.

We were of course very cognizant of matching controls in this study. Because these are human participants and not mice, we cannot match them on all fronts. Thus, we are forced to rank-order the importance of the matching criteria, and maximally match on the most important fronts, at the risk of variability on other counts. Our two most important matching criteria were: 1. Age, 2. Number of living children and their age. Beyond that, we always tried to match means on all other fronts. In other words, even if we did have individual differences on lower-order considerations, such as living arrangements, these differences were not systematically associated with group (uRPL vs. Control).

We further note that as to: "groups may differ in the number of total pregnancies", this is of course the case by definition. Although we are not aware of any study suggesting that number of pregnancies influences olfaction, this is of course a possibility, but this difference is inherent to our study. Given equal number of live children, the uRPL group will always have more pregnancies. To ask if number of pregnancies is related to these results, what we can do is to take the most prominent behavioral result we have: spouse identification, and see if it is associated with number of pregnancies within each group (collapsing across both groups would be biased since RPL women were better at the task and, as said, have higher number of pregnancies). If the correlation reflects a relation of olfaction and pregnancies, it should materialize within the groups as well. Doing so we find no significant correlations. This control analysis has been added to the text. Regarding number of sexual partners and living arrangements, all women in the identification task, and the overwhelming majority of women in the experiment, were married and living with their spouses at time of the experiment. We now note this in the Materials and methods as well:

“Because this study involves natural groups rather than randomization, we paid close attention to controlling for factors which may confound our findings. For each comparison performed, we matched control participants to uRPL participants based on their age and number of children (see Tables 1-4 for details). If multiple choices were available, our choice was based on matching the average of the control group as a whole. For the olfactory bulb analysis, following recent findings regarding the impact of pregnancy on maternal brain anatomy after birth (Kim et al., 2010; Hoekzema et al., 2017; Kim et al., 2018), we additionally matched for age of offspring, particularly age of the youngest child (Table 3). Additional factors, like marital status and living arrangements, were similar between groups: the overwhelming majority of women in our study were married and living with their spouses. One factor that is by definition confounded with group differences is number of pregnancies, as RPL is defined as 2 or more consecutive miscarriages. To ask whether number of pregnancies may be associated with our results, we examined the most prominent behavioral result, namely spouse identification, and its association with number of pregnancies within each group. We observed no significant correlations (RPL group: Spearman’s Rho = 0.19, p = 0.3; Control group: Spearman’s Rho = -0.29, p = 0.11), implying that number of pregnancies alone does not impact this result."

• A curious finding is the negative correlation between the composite olfactory score and the spouse identification performance in the RPL group. Would we not expect a positive correlation and importantly present across all participants?

First, we would like to caution that this negative correlation was not statistically significant, so we would not characterize it as a "finding". Nevertheless, a trend is there, and it is counter-intuitive. Indeed, we would expect a positive correlation, and that was certainly not there. Although we could hand-wave, we have absolutely no good explanation for this. Thus, all we can do is make sure to clearly note this, and retain it as an open observation. We now stress this in the manuscript.

Reviewer #1:This manuscript describes a series of experiments testing for "Bruce effect"-like effects in humans. The key findings are that compared to controls, human females with repeated pregnancy loss (RPL) were able to identify the body odor of their spouses, despite relatively comparable olfactory perceptual ability. Compared to controls, these women rated stranger's body odors differently and also showed increased activity in the hypothalamus to such odors. In addition, women with RPL had significantly smaller olfactory bulbs.This is a expertly conducted study on an important topic with interesting results. The findings are important because they may lead to new interventions for RPL. That said, there are a few issues that should be considered.1) The effects for the spouse-odor identification test are strong and convincing. However, the results of the follow-up tests, aimed at identifying the specific cause of this advantage, are much weaker and less conclusive. From this perspective, it is less clear that women with RPL have an heightened sense of smell, and thus these findings should be interpreted with more caution. If there are no clear differences between groups, this opens up the possibility of alternative explanations. For instance, it is tempting to speculate that non-peripheral mechanisms, likely downstream of olfactory bulb are at play. Such effects could for instance be related to improved odor memory.

We completely agree, and this is in fact the outcome we were trying to convey. If there is any overall olfactory advantage in RPL, it is very minimal. We just didn't want to be in the place of even remotely obscuring such an advantage, as a claim that the RPL advantage is limited to only social odors, although consistent with our initial hypothesis, may be a stretch. Nevertheless, we now better clarify this in the Discussion.

2) Alternatively, it is possible that the effects for spouse-odor identification are not at all related to RPL women's olfactory system but that they reside in their partners. As acknowledged by the authors, the men in RPL relationships may have unique body odors that are easy to be identified. One way to rule this out is to use the same set of odors in both groups, such that the spouse odor in one group is the non-spouse odor in the other. If the effects are driven by distinctive spouse-odors in the RPL group, women in both groups should be able to correctly identify this odor resulting in similar performance across groups. That is, the RPL group could correctly identify their spouse in the RPL group, and the control group could correctly identify these stand-out odors as the non-spouse odor. Ideally, the authors would rule out this important possibility in a control experiment. At the very least, however, this should be acknowledged and discussed. This confound is absent in many of the other experiments including the imaging studies. So while this potential explanation of the spouse-odor identification results needs to be taken seriously, it does not distract from the overall importance of this study.

We completely concur. Beyond comments in the results, we now have a section in the Discussion addressing both this and the previous comment:

"We would like to reiterate two considerations related to this primary behavioral finding. First, considering the only marginal uRPL advantage at other olfactory tasks, this advantage may reflect some downstream uRPL advantage, perhaps even memory beyond the olfactory system. Second, as mentioned in the results, this advantage may reflect not better olfactory memory/identification in uRPL women, but rather a unique body-odor in uRPL men. Alas, our experimental design did not permit addressing this question in the current results, and this remains an open and tantalizing question for future research. That said, if there is a body-odor difference in uRPL men, we speculate that this is in addition and not instead of altered body-odor perception in uRPL women. We say this because in addition to better performance in spouse identification, uRPL women also reported different perceptual qualities associated with a group of independent men. Women with and without uRPL rated the same group of non-spouse men, and the women with RPL perceived them differently (Figure 2). Thus, we stand by the notion that women with uRPL have altered perception of men's body-odor."

3) Because this study involves natural groups rather than randomization, there are a number of important confounds that need to be controlled for. For instance, groups may differ in the number of total pregnancies and other factors that are related to RPL. These factory could have an effect on olfaction in RPL women or on their ability to recognize their spouse's odor. Moreover, other factors may include the number of sexual partners, their living arrangements, etc. These and other factors could explain the present association and this should be tested as possible. In general, more needs to be said in the main text about how the groups were matched and how potential confounds were ruled out.

As noted in the reply to Editors: We were of course very cognizant of matching controls in this study. Because these are human participants and not mice, we cannot match them on all fronts. Thus, we are forced to rank-order the importance of the matching criteria, and maximally match on the most important fronts, at the risk of variability on other counts. Our two most important matching criteria were: 1. Age, 2. Number of living children and their age. Beyond that, we always tried to match means on all other fronts. In other words, even if we did have individual differences on lower-order considerations, such as living arrangements, these differences were not systematically associated with group (uRPL vs. Control).

We further note that as to: "groups may differ in the number of total pregnancies", this is of course the case by definition. Although we are not aware of any study suggesting that number of pregnancies influences olfaction, this is of course a possibility, but this difference is inherent to our study. Given equal number of live children, the uRPL group will always have more pregnancies. To ask if number of pregnancies is related to these results, what we can do is to take the most prominent behavioral result we have: spouse identification, and see if it is associated with number of pregnancies within each group (collapsing across both groups would be biased since RPL women were better at the task and, as said, have higher number of pregnancies). If the correlation reflects a relation of olfaction and pregnancies, it should materialize within the groups as well. Doing so we find no significant correlations (both p > 0.1). This control analysis has been added to the text. Regarding number of sexual partners and living arrangements, all women in the identification task, and the overwhelming majority of women in the experiment, were married and living with their spouses at time of the experiment. We now note this in the Materials and methods as well:

“Because this study involves natural groups rather than randomization, we paid close attention to controlling for factors which may confound our findings. For each comparison performed, we matched control participants to uRPL participants based on their age and number of children (see Tables 1-4 for details). If multiple choices were available, our choice was based on matching the average of the control group as a whole. For the olfactory bulb analysis, following recent findings regarding the impact of pregnancy on maternal brain anatomy after birth (Kim et al., 2010; Hoekzema et al., 2017; Kim et al., 2018), we additionally matched for age of offspring, particularly age of the youngest child (Table 3). Additional factors, like marital status and living arrangements, were similar between groups: the overwhelming majority of women in our study were married and living with their spouses. One factor that is by definition confounded with group differences is number of pregnancies, as RPL is defined as 2 or more consecutive miscarriages. To ask whether number of pregnancies may be associated with our results, we examined the most prominent behavioral result, namely spouse identification, and its association with number of pregnancies within each group. We observed no significant correlations (RPL group: Spearman’s Rho = 0.19, p = 0.3; Control group: Spearman’s Rho = -0.29, p = 0.11), implying that number of pregnancies alone does not impact this result."

4) A curious finding is the negative correlation between the composite olfactory score and the spouse identification performance in the RPL group. Would we not expect a positive correlation and importantly present across all participants?

As noted in the reply to Editors: First, we would like to caution that this negative correlation was not statistically significant, so we would not characterize it as a "finding". Nevertheless, a trend is there, and it is counter-intuitive. Indeed, we would expect a positive correlation, and that was certainly not there. Although we could hand-wave, we have absolutely no good explanation for this. Thus, all we can do is make sure to clearly note this, and retain it as an open observation. We now stress this in the manuscript.

Reviewer #2:Rozenkrantz et al. investigate links between repeated pregnancy loss (RPL) and women's perception of male body odor, hypothesizing an underlying Bruce-like effect. The authors find several perceptual and neuroanatomical differences between RPL and control groups. These effects are certainly intriguing, especially the reported higher ability of RPL women to identify their spouses, as well as the neuroanatomical differences between this group and controls. However, these data do currently not allow to conclude a Bruce-like effect in humans. In my opinion, this study would therefore only be publishable in eLife conditional on addressing the concerns outlined below.1) Extensive parallels are drawn to the Bruce effect, the mechanisms of which have most extensively been described in rodents. In mice, pheromone detection via the vomeronasal pathway and subsequent hypothalamic activation and dopamine release results in a loss of pregnancy only prior to implantation. In contrast, vomeronasal signalling is likely absent in humans and miscarriages (or pregnancies for that matter) would only be diagnosed long after embryo implantation. Even though the authors explicitly state that the Bruce effect is unlikely to be present in humans, they extensively refer to this phenomenon and e.g. place the hypothalamic activation in RPL women in this context. Although this limited equivalence is being addressed in the Discussion, allusions to a Bruce(-like) effect should feature less prominently throughout the Abstract/Introduction and Results sections.

We embrace this criticism, and have deleted all analogies with the Bruce effect from the Abstract, Introduction, and Results, and now retain only a very cautious consideration of this at the very final paragraph of the Discussion alone.

2) The authors' use targeted analysis of their structural and functional imaging data to identify differences (OB volume, hypothalamic activation) between RPL women and controls. Yet, when the same data are analyzed in a hypothesis-free manner, the same differences cannot be identified (apart from when data are not corrected for multiple comparisons). This should be discussed.

As noted in the reply to Editors: We acknowledge this, but stress that it is in no way an anomaly. For example, previous studies that uncover a hypothalamic response to Androstadienone were all based on region of interest (ROI) analyses:

https://www.ncbi.nlm.nih.gov/pmc/articles/PMC3397979/ https://www.ncbi.nlm.nih.gov/pmc/articles/PMC1129091/ https://www.ncbi.nlm.nih.gov/pmc/articles/PMC4037295/ Nevertheless, we now further stress this in the Discussion.

3) A Bruce-like effect would rely on the correct identification of a smell as belonging to a Stranger. The study design chosen here however asked participants to distinguish between Spouse, Stranger and blank. While the authors argue that the correct identification of a familiar Spouse smell is a prerequisite for identifying other smells as Stranger, I am curious why participants were not asked to choose between e.g. two (or several) Spouse jars and one Stranger jar.

As noted in the reply to Editors: As our initial intentions were to test whether RPL women were better able to recognize their spouse’s body odor than control women, the task we designed was an identification task. Had we used the design suggested here, we would have been testing their discrimination abilities, that is: given two identical odors and one different odor, are they able to tell the different odor apart. Thus, we constructed the task in accordance with our motivation to examine spouse recognition.

Reviewer #3:In "Unexplained Repeated Pregnancy Loss is Associated with Altered Perceptual and Brain Responses to Men's Body-Odor" Rozenkrantz et al. explore the hypothesis that repeated pregnancy loss (RPL) in humans may be olfactory driven, similar to the Bruce effect in mice. When assayed for the ability of women to guess the odor of their spouse verses another male, they find the population of women that experienced RPL to perform better than women that were easily able to conceive. The correlation is extended to anatomical differences in the size of RPL olfactory bulbs and functional activity differences to male odors assayed by fMRI. Overall, the data is correlative, as is expected with human studies, there is no direct evidence that olfaction is contributing to RPL. As I read the manuscript, I tallied a list of major concerns, and then found them largely clearly stated in the last two paragraphs of the Discussion. The Bruce effect is only superficially analogous to the observations and experiments here, and the claims are much more limited than what is implied in the Abstract. With such a devastating condition as RPL, I do not think the story should be speculatively comingled with a unique aspect of behavior of a subset of species. I recommend removing most references to the Bruce effect prior to publication.

We embrace this criticism. As noted in the reply to Editors, we have removed all mention of the Bruce effect from the Abstract, Introduction and Results, and have limited its consideration to a careful final comment in the Discussion.

1) Though there is a statistical difference between RPL and control in Figures 1A/2/3B the response of both populations are very broad and the overall spread is very similar between RPL and control women. This looks like an example of a coincidental statistical difference – but not a biologically meaningful difference. I expect that if one was to look across all women (impossible) the results would not hold. With such a large variance in both populations in all measures of study, I do not think the data supports the conclusions.

This is a keen observation, and to address this possibility we conducted bootstrap analyses for all three results. In each case, we randomly shuffled the assignment of the data to uRPL or Control 10,000 times, and reconducted the analysis. We observe that in all three cases, our result is far removed from chance. More specifically, the probability of obtaining these results coincidentally is 2 in 1000 for Figure 1A, 6 in 1000 for Figure 2, and 2 in 100 for Figure 3B. Based on this analysis, we think the results are quite robust. We have added this important control analysis to all the figures.

2) The Materials and methods indicate that for the experiments presented in Figures 1-2 each woman sniffed the SSD either 4-6 times. It is not clear why some women underwent two additional attempts. The “individual” data points represented in the figures are an average of these 4-6 trials. Please include a table of the raw data points for each individual to reveal the variance in response and identify the conditions where the additional trials occurred.

As noted in the reply to Editors: Some participants had different numbers of trials for the following reason: This study was conducted over many years (the recruitment rate was slow), and naturally, we evolved some of the paradigms on the way, in order to improve them. In this case, it dawned on us that it would be more powerful to be able to assess significance also within each participant alone and not only across groups, so we added trials. That said, critically, the number of trials remained equal across cohorts. In other words, for every RPL participant with a reduced number or trials there was a matched control with equally reduced trials. Thus, this shortcoming cannot underlie the group effect that we report. Moreover, in consideration of this comment, we now tested again, using only the first four trials from all those participants that had 6 trials, and we obtain ostensibly the same results. Finally, there were 3 cases where we had only two trials. If we restrict the entire cohort to just two trials, the effect again still remains significant. To conclude, the group effect we reported was not influenced by the number of trials conducted per participant. Finally, consistent with the reviewer suggestion, we have now included a table (Supplementary file 2) which details all trials for each participant, and we added these sub-analyses.

3) Was the identity of the odor (spouse or stranger) blind to the researcher that presented the SSDs to each individual (I cannot find this in the Materials and methods)? If not, subliminal cues by the experimenter could alter the individual's perception and should be stated as a potential reason for behavioral differences observed.

As noted in the reply to Editors: This experimental design was not double-blind in the strictest sense of the term, but its arrangement largely prevented the risk of experimenter-generated cues. More specifically, during each trial the experimenter did not hand the participant one SSD at a time, but rather placed three arbitrarily marked SSDs on the table in front of the participant at the beginning of the trial. The experimenter then moved away, and the participant alone interacted with a computer that provided sampling instructions. The participant then entered their selection using the trackpad of the computer, such that the experimenter did not know what the participant answered at that time. There was no voice interaction between the experimenter and participant during a trial. In other words, the timing of each sniff, its content, and answer, were all independently controlled by the participant without experimenter involvement. Nevertheless, we acknowledge that this is not genuinely double-blind, and moreover, we acknowledge that this is a drawback. Indeed, in retrospect, an unnecessary drawback.

4) The functional fMRI differences are confounded by watching video clips during assay. Why is this necessary? Of relevance here is the activity response towards odors only. The conclusions would be stronger if the stimuli were clear and controlled.

As noted in the reply to Editors: We agree that the direct response to odor alone is an interesting path to take. That said, we opted to go for modulation of activity related to arousal because of a set of considerations that we think are strong. Specifically: uRPL participants are a "valuable" participant type. That is, we cannot recruit endless participants as we could Controls, and therefore, we wanted to maximize chances of obtaining meaningful signal in our first shot. A subliminal odor alone drives very minimal patterns of activity, but is far more evident in the brain in its modulation of other, related task activations. This notion of social odors as a human brain "modulator" rather than brain "activator" has been well validated (Jacob, Hayreh and McClintock, 2001). Of course, had our interest been the nature of response to body-odors alone, this would have been more of a problem, but given that our interest is the difference between groups as a function of body-odor, we judged this to be the correct path to take. In light of this comment, we have added text to justify this path in the manuscript.

Reviewer #4:The data that you present are of high quality and may be of great value to the RPL/medical literature, although I am unable to assess that aspect of your paper in detail. My reviewing expertise lies in the connection between your manuscript and the Bruce effect literature, and I believe that you would be better off avoiding discussion of the Bruce effect entirely. It is unclear that there is or should be any mechanistic or functional connection between RPL in humans and the Bruce effect, and I fear the addition of this manuscript to the Bruce effect literature would do more to confuse than to clarify.I don't wish to belabour the below points more than I already have. In areas where my explanation may lack detail, I would recommend a thorough read ofdeCatanzaro (2015) "Sex steroids as pheromones in mammals: the exceptional role of estradiol." Hormones and Behavior.andZipple et al. (2019) Male-mediated prenatal loss: Functions and mechanisms." Evolutionary Anthropology.

We have now read both, and thank you for directing us in this

1) The mechanistic argument: In order for use of the term "Bruce effect" or even "Bruce-like effect" to be justified, it seems necessary to show that there is some reason to believe that terminations of pregnancies are occurring after women are exposed to non-sire males.You attempt to make this connection by showing that, compared to control women, women that display RPL are (1) more discriminating in their olfaction, especially their ability to distinguish between a spouse and a non-spouse, (2) have smaller olfactory bulbs, and (3) show differential brain response to men's body odors. The underlying logic appears to be that women who display RPL are sensitive to the smell of other men, and after smelling a non-spouse man, a chemical cascade is triggered that results in the termination of their pregnancies.Such a line of argumentation would be much firmer if we had reason to believe that exposure to the odors of non-sire males is the trigger that leads to pregnancy termination across the many species that exhibit the Bruce effect, including some primates. Under such a scenario the lack of causality in the paper might justify the speculation that the presented data support a Bruce effect in humans. However, at present there is no evidence in any species that pregnancy termination (as opposed to implantation failure) is triggered by exposure to the scents of non-sire males. Even in mice, where the odor of non-sire males has been argued to trigger a Bruce effect by preventing implantation, it is unclear that exposure to the odor alone is sufficient to trigger the Bruce effect in the absence of exogenous estradiol.What's more, it seems unlikely that such a complex and costly behavior of pregnancy termination could be triggered in humans simply as a result of a conspecific chemical cue, especially in a species in which females routinely encounter non-spouse males. Outside of rodents, there has so far been no evidence that odor or any other exogenous chemical mechanism is at play in species that exhibit the Bruce effect, such as dogs, horses, or primates.While it is possible that olfactory cues play a role in post-implantation Bruce effects, it is also possible that the relevant cues in these cases are entirely social/visual and require no chemical cue whatsoever. Animals that are capable of recognizing each other visually, both because they have the cognitive capacity for such an assessment and because they live in a visual environment that allows for it, seem much more likely to rely on sociocognitive rather than exogenous chemical cues to trigger termination. A woman (or for that matter a female horse, dog, or lion) does not need to smell a non-sire male to know that he is present in her social group nor does she need to smell a male to know that he might pose a threat to the survival of her future children. Rather, she is able to assess changes in the dynamics of her group to make such determinations in a much more accurate way.So, from a mechanistic standpoint, the speculation about associations between olfactory sensitivity and RPL does not appear justified.

Although we feel slightly odd to here provide a short answer to this very detailed and long concern, we can briefly say that we accepted and embraced this point of view, and therefore completely reframed our manuscript, deleting nearly all reference to the Bruce effect from the Abstract, Introduction, and body, and retaining only minimal allusion to it in the Discussion, as suggested by the Editors.

2) The evolutionary argument: I also have concerns about the evolutionary story that you are telling here. Your argument appears to be that the ability of control women not to miscarry following exposure to the scent of non-sire men is an adaptation to communal living. The implication appears to be that termination following exposure to the scent of non-sire males is the ancestral state in primates, a claim that seems unlikely given the above discussion of mechanisms. It is also a confusing claim, because a great many primates live "communally," but there are certainly specie in which the Bruce effect does not occur (e.g. chacma baboons and yellow baboons). So when did this adaptation occur? Was it a prerequisite for group-living? If so, then it must have occurred millions of years ago.If we take this interpretation as accurate, the unstated evolutionary argument that you are making is that the ancestral chemical cascade that had previously induced termination following exposure to non-sire males was broken via a mutation in an olfactory receptor gene (OR4C16G>A). For millions of years since that cascade was broken, it has remained otherwise entirely functional except that this change in one olfactory reception gene has caused it to no longer be triggered. So, when women are born with a mutation in the olfactory reception gene that makes it functional once more, the necessary chemical pathways that lead from exposure to male odors to pregnancy termination kick in and continue to work just as they would have millions of years previously. This seems to me to be a very unlikely scenario. There are additional issues that I see in terms of the evolutionary narrative at play, but I will not belabor these points further.I therefore see no reason to refer to the Bruce effect in this manuscript, except as perhaps a passing comment in the Discussion that suggests it is possible that there may be some connection. Even this feels like more speculation than may be warranted.

Again, we embraced this point of view, and radically edited the manuscript to match.

Reviewer #5:This manuscript describes experiments suggested by the argument that human females exhibit a Bruce Effect in which pregnant women spontaneously abort gestation when exposed to the odor of men other those who impregnated them. The experiments involve exposing women with and without histories of repeated pregnancy loss to odors of partners as well as other men to determine, in general, whether those with such histories detect partner odor better than controls, whether their olfactory biology differs from controls, and whether they react differently to arousing stimuli when exposed to male body odors. The pattern of results support the argument that women with a history of repeated pregnancy loss appear different from controls in way consistent with a Bruce Effect in women.Those of us inclined to believe that humans exhibit Bruce Effect like behavior will find your arguments and findings powerful. Skeptics will offer caveats many of which you anticipated, and both directly and indirectly defend against. But to further avoid capitalizing on confirmation bias among readers, I suggest you alert them to two circumstances. First, the literature that explains, rather than describes, the "Bruce Effect" characterizes the termination of gestation when the environment threatens the survival of offspring as adaptive. Natural selection presumably conserved mutations that produce the Effect because they increase the reproductive fitness of females. They should, therefore, appear frequently among females in a stable, self-sustaining population. You, however, study a clinically defined sub-group of the population. A sub-group that exhibits a non-adaptive behavior, very frequent pregnancy loss. Your findings could be characterized as identifying risk factors for inclusion in a relatively small group that does not contribute to a self-sustaining population (i.e., 2-4% of all women who suffer frequent clinically recognized fetal loss), rather than providing evidence for an adaptive mechanism that, if the Bruce Effect argument is correct, should appear in a large fraction of women. You need, in short, to discuss the external validity of your findings and its implications for theory.

Put simply: we could not agree more, and although this is a document where the reviewer is commenting on our work, and not the reverse :-), we would nevertheless like to say that the above is very well put. That said, because the review consensus was that we remove most all reference to the Bruce effect, and we grudgingly accepted this, all this becomes rather moot. The Bruce effect is now largely out of this manuscript. We will soon set out on a line of added experimentation that if successful, will imply causation, and if we get there, we would like to use the reviewer comment verbatim.

Second, you should spend a bit of page space making clear that the literature includes tests of a generalized Bruce Effect in humans. That work shows that, consistent with the theory underlying the Bruce Effect, populations witnessing unexpected death among children suffer greater than expected male fetal loss. The work remains controversial because it invokes "Bruce Effect" to describe the termination of gestation following something other than exposure to the odor of men. But the work does allow you to argue that your reasoning and findings may apply to circumstances other than rare cases of repeated pregnancy loss.

Although, as noted, the Bruce effect has been largely stripped from this manuscript, we now address this perspective in the Discussion.